# Preservation of the Global Knowledge by Not-True Distillation in Federated Learning

**Gihun Lee\*, Minchan Jeong\*, Yongjin Shin, Sangmin Bae, Se-Young Yun**
KAIST
{opcrisis, mcjeong, yj.shin, bsmn0223, yunseyoung}@kaist.ac.kr

## Abstract

In federated learning, a strong global model is collaboratively learned by aggregating clients' locally trained models. Although this precludes the need to access clients' data directly, the global model's convergence often suffers from data heterogeneity. This study starts from an analogy to continual learning and suggests that *forgetting* could be the bottleneck of federated learning. We observe that the global model forgets the knowledge from previous rounds, and the local training induces forgetting the knowledge outside of the local distribution. Based on our findings, we hypothesize that tackling down forgetting will relieve the data heterogeneity problem. To this end, we propose a novel and effective algorithm, *Federated Not-True Distillation* (FedNTD), which preserves the global perspective on locally available data only for the *not-true* classes. In the experiments, FedNTD shows state-of-the-art performance on various setups without compromising data privacy or incurring additional communication costs[1].

## 1 Introduction

At present, massive data is being collected from edge devices such as mobile phones, vehicles, and facilities. As the data may be distributed on numerous devices, decentralized training is often required to train deep network models. Federated learning [23, 24] is a distributed learning paradigm that enables the learning of a global model while preserving clients' data privacy. In federated learning, clients independently train local models using their private data, and the server aggregates them into a single global model. In this process, most of the computation is performed by client devices, while the global server only aggregates the model parameters and distributes them to clients [2, 50].

Most federated learning algorithms are based on FedAvg [37], which aggregates the locally trained model parameters by weighted averaging proportional to the amount of local data that each client had. While various federated learning algorithms have been proposed thus far, they each conduct parameter averaging in a certain manner [1, 9, 20, 30, 47, 52]. Although this aggregation scheme empirically works well and provides a conceptually ideal framework when all client devices are active and i.i.d. distributed (a.k.a. LocalSGD), the data heterogeneity problem [31, 56] is a notorious challenge for federated learning applications and prevents their widespread applicability [19, 29].

As the clients generate their own data, the data is not identically distributed. More precisely, the local data across clients are drawn from heterogeneous underlying distributions; thereby, locally available data fail to represent the overall global distribution, which is referred to as data heterogeneity. Despite its inevitable occurrence in many real-world scenarios, data heterogeneity not only makes theoretical analysis difficult [31, 56] but also degrades many federated learning algorithms' performances [18, 27]. By resolving the data heterogeneity problem, learning becomes more robust against partial participation [31, 37], and the communication cost is also reduced by faster convergence [46, 52].

---

\* Equal contribution.
[1] https://github.com/Lee-Gihun/FedNTD

36th Conference on Neural Information Processing Systems (NeurIPS 2022).

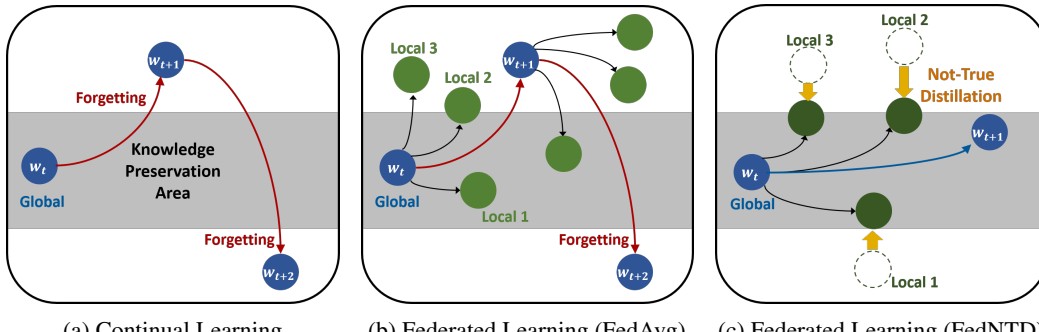

Figure 1: An overview of forgetting in learning scenarios. As catastrophic forgetting in (a) Continual Learning, (b) Federated Learning also experiences forgetting. However, (c) FedNTD prevents forgetting by preserving global knowledge during local training.

Interestingly, continual learning [42, 45] faces a similar challenge. In continual learning, a learner model is continuously updated on a sequence of tasks, with an objective to perform well on whole tasks. Unfortunately, owing to the heterogeneous data distribution of each task, learning on the task sequence often results in *catastrophic forgetting* [36, 40], whereby fitting on a new task interferes with the parameters important for previous tasks. As a consequence, the model parameters drift away from the area where the previous knowledge is desirably preserved (Figure 1a).

Our first conjecture is such forgetting also exists in federated learning. While the server aggregates local models, the distribution where they are trained may be largely different from those of previous rounds. As a result, the global model faces the distributional shifts at each round, which may cause the forgetting as in continual learning (Figure 1b). To empirically verify this analogy, we examine the global model's prediction consistency. More specifically, we measure its class-wise accuracy while the communication rounds proceed.

The observations verify our conjecture: the global model's prediction is highly inconsistent across communication rounds, significantly reducing the performance of predicting some classes that the previous model originally predicted well. We dig deeper to analyze how averaging the locally updated parameters induces such forgetting and confirm that it occurs in local training: the global knowledge, corresponding to the region outside of local distribution, is prone to be forgotten. As merely averaging the local models cannot recover it, the global model struggles to preserve previous knowledge.

Based on our findings, we hypothesize that mitigating the issue of forgetting can relieve data heterogeneity (Figure 1c). To this end, we propose a novel algorithm Federated Not-True Distillation (FedNTD). FedNTD utilizes the global model's prediction on locally available data, but only for the not-true classes. We demonstrate the effect of FedNTD on preserving global knowledge outside of a local distribution and its benefits on federated learning. Experimental results show that FedNTD achieves state-of-the-art performance in various setups.

To summarize, our contributions as follows:

- We present a systematic study on forgetting in federated learning. The global knowledge outside of the local distribution is prone to be forgotten and is closely related to the data heterogeneity issue **(Section 2)**.

- We propose a simple yet effective algorithm, FedNTD, to prevent forgetting. Unlike prior works, FedNTD neither compromises data privacy nor incurs additional communication burdens. We validate the efficacy of FedNTD on various setups and show that it consistently achieves state-of-the-art performance **(Section 3, Section 4)**.

- We analyze how FedNTD benefits federated learning. The knowledge preservation by FedNTD improves weight alignment and weight divergence after local training **(Section 5)**.

### 1.1 Preliminaries

**Federated Learning** We aim to train an image classification model in a federated learning system that consists of $K$ clients and a central server. Each client $k$ has a local dataset $\mathcal{D}^k$, where the whole dataset $\mathcal{D} = \bigcup_{k \in [K]} \mathcal{D}^k$. At each communication round $t$, the server distributes the current global

model parameters $w^{(t-1)}$ to sampled local clients $K^{(t)}$. Starting from $w^{(t-1)}$, each client $k \in K^{(t)}$ updates the model parameters $w_k^{(t)}$ using its local datasets $\mathcal{D}^k$ with the following objective:

$$w_k^{(t)} = \underset{w}{\operatorname{argmin}} \ \mathbb{E}_{(x,y) \sim \mathcal{D}^k}[\mathcal{L}(w; w^{(t-1)}, x, y)]. \tag{1}$$

where $\mathcal{L}$ is the loss function. At the end of round $t$, the sampled clients upload the locally updated parameters back to the server and aggregate by parameter averaging as $w^{(t)}$ as follows:

$$w^{(t)} = \sum_{k \in K^{(t)}} \frac{|\mathcal{D}^k|}{\sum_{k' \in K^{(t)}} |\mathcal{D}^{k'}|} \ w_k^{(t)}. \tag{2}$$

**Knowledge Distillation**   Given a teacher model $T$ and a student model $S$, knowledge distillation [17] matches their softened probability $q_\tau^T$ and $q_\tau^S$ using temperature $\tau$. The $c$-th value of the $q_\tau$ can be described as $q_\tau(c) = \frac{\exp(z_c/\tau)}{\sum_i \exp(z_i/\tau)}$, where $z_c$ is the $c$-th value of logits vector $z$ and $\mathcal{C}$ is the number of classes. Given a sample $x$, the student model $S$ is learned by a linear combination of cross-entropy loss $\mathcal{L}_{\text{CE}}$ for one-hot label $\mathbb{1}_y$ and Kullback-Leibler divergence loss $\mathcal{L}_{\text{KL}}$ using a hyperparameter $\beta$:

$$\mathcal{L} = (1 - \beta)\mathcal{L}_{\text{CE}}(q, \mathbb{1}_y) + \beta\tau^2 \mathcal{L}_{\text{KL}}(q_\tau^S, q_\tau^T) \tag{3}$$

$$\mathcal{L}_{\text{CE}}(q, \mathbb{1}_y) = -\sum_{c=1}^{\mathcal{C}} \mathbb{1}_y(c) \log q(c), \qquad \mathcal{L}_{\text{KL}}(q_\tau^S, q_\tau^T) = -\sum_{c=1}^{\mathcal{C}} q_\tau^T(c) \log \left[ \frac{q_\tau^S(c)}{q_\tau^T(c)} \right] \tag{4}$$

## 2   Forgetting in Federated Learning

To understand how the non-IID data affects federated learning, we performed an experimental study on heterogeneous locals. We choose CIFAR-10 [25] and a convolutional neural network with four layers as in [37]. We split the data to 100 clients using Latent Dirichlet Allocation (LDA), assigning the partition of class $c$ samples to clients by $p \sim \text{Dir}(\alpha)$. The heterogeneity level increases as the $\alpha$ decreases. We train the model with FedAvg for 200 communication rounds, and 10 randomly sampled clients are optimized for 5 local epochs at each round. More details are in Appendix B.

### 2.1   Global Model Prediction Consistency

To confirm our conjecture on forgetting, we first consider how the global model's prediction varies as the communication rounds proceed. If the data heterogeneity induces forgetting, the prediction after update (i.e., parameter averaging) may be less consistent compared to the previous round. To examine it, we observe the model's class-wise test accuracy at each round, and measure its similarity to the previous round. The results are provided in Figure 2a and Figure 2b.

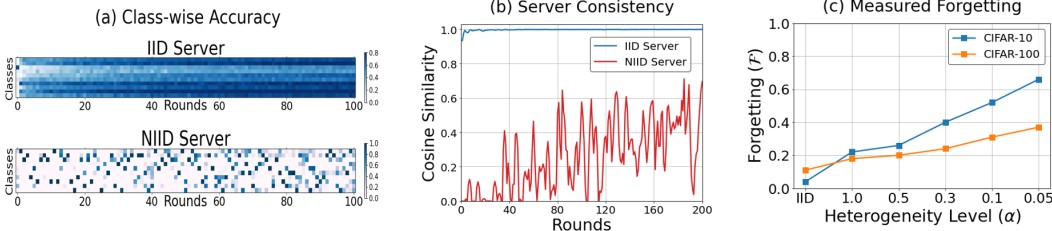

Figure 2: Forgetting analysis on the global server model. (a): Class-wise test accuracy on CIFAR-10 IID and NIID ($\alpha$=0.1) cases. (b): Cosine similarity of class-wise accuracy vector w.r.t. previous round global model on IID and NIID ($\alpha = 0.1$) cases. (c): Forgetting $\mathcal{F}$ by different heterogeneity levels on CIFAR-10 and CIFAR-100.

As expected, while the server model learned from i.i.d. locals (IID Server) predicts each class evenly well at each round, prediction is highly inconsistent in the non-i.i.d. case (NIID Server). In the non-IID case, the test accuracy on some classes which originally predicted well by the previous global model often drops significantly. This implies that forgetting occurs in federated learning.

To measure how the forgetting is related to data heterogeneity, we borrow the idea of *Backward Transfer* (BwT) [5], a prevalent forgetting measure in continual learning [4, 7, 8, 14], as follows:

$$\mathcal{F} = \frac{1}{\mathcal{C}} \sum_{c=1}^{\mathcal{C}} \max_{t \in \{1, \dots \mathcal{T}-1\}} (\mathcal{A}_c^{(t)} - \mathcal{A}_c^{(\mathcal{T})}) \tag{5}$$

where $\mathcal{A}_c^{(t)}$ is the accuracy on class $c$ at round $t$. Note that the forgetting measure, $\mathcal{F}$, captures the averaged gap between peak accuracy and the final accuracy for each class at the end of learning. The result on the varying heterogeneity levels is plotted in Figure 2c, showing that the global model suffers from forgetting more severely as the heterogeneity level increases.

## 2.2 Knowledge Outside of Local Distribution

We take a closer look at local training to investigate why aggregating the local models induces forgetting. In the continual learning view, a straightforward approach is to observe how fitting on the new distribution degrades the performance of the old distribution. However, in our problem setting, the local clients can have any class. Given that only their portions in the local distribution differ across clients, such strict comparison is intractable. Hence, we formulate *in-local distribution* $p(\mathcal{D})$ and its *out-local distribution* $\tilde{p}(\mathcal{D})$ to systematically analyze forgetting in local training.

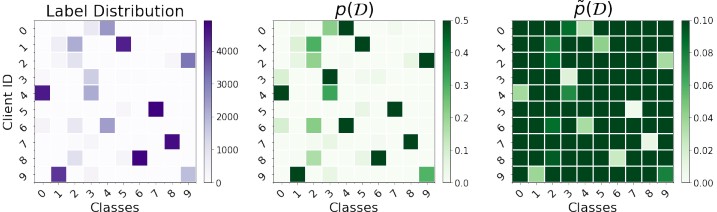

Figure 3: An example of in-local distribution $p(\mathcal{D})$ and out-local distribution $\tilde{p}(\mathcal{D})$ on CIFAR-10 ($\alpha = 0.1$).

**Definition 1.** *Consider a local dataset $\mathcal{D}$ consists of $N$ data points $x_i$ and its label $y_i$ in $\mathcal{C}$-class classification problem. The **in-local distribution vector** $p^k = p(\mathcal{D}^k)$ and its **out-local distribution vector** $\tilde{p}^k = \tilde{p}(\mathcal{D}^k)$ are*

$$p = [p_1, \ldots, p_{\mathcal{C}}], \quad \text{where} \quad p_c := \frac{1}{N} \sum_{i=1}^{N} \mathbb{I}(y_i = c) \tag{6}$$

$$\tilde{p} = [\tilde{p}_1, \ldots, \tilde{p}_{\mathcal{C}}], \quad \text{where} \quad \tilde{p}_c := \frac{1}{\mathcal{C} - 1}(1 - p_c) \tag{7}$$

The underlying idea of out-local distribution $\tilde{p}(\mathcal{D})$ is to assign a higher proportion to the classes with fewer samples in local datasets. Accordingly, it corresponds to the region in the global distribution where the in-local distribution $p(\mathcal{D})$ cannot represent. Note that if $p(\mathcal{D})$ is uniform, $\tilde{p}(\mathcal{D})$ also collapses to uniform, which aligns well intuitively. An example of label distribution for 10 clients and their $p(\mathcal{D})$s and $\tilde{p}(\mathcal{D})$s are provided in Figure 3.

We measure the change of global and local models' accuracy on $p(\mathcal{D})$ and $\tilde{p}(\mathcal{D})$ during each communication round, as in Figure 4. After local training, the local models are well-fitted towards $p(\mathcal{D})$ (Figure 4a), and the aggregated global model also performs well on it. On the other hand, the accuracy on $\tilde{p}(\mathcal{D})$ significantly drops, and the global model accuracy on it also degrades (Figure 4b).

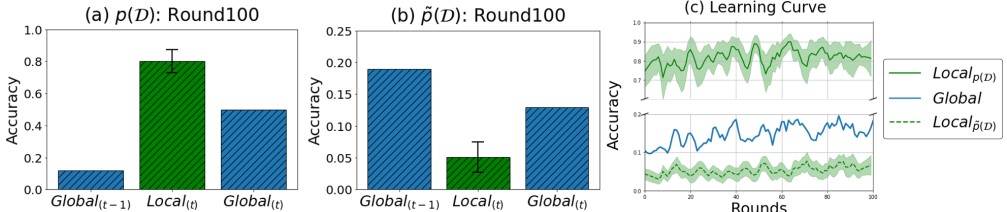

Figure 4: Accuracy of global model and sampled local models for $p(\mathcal{D})$ and $\tilde{p}(\mathcal{D})$ on CIFAR-10 ($\alpha$=0.1). The error bar stands for the standard deviation of the 10 sampled local clients. In (a) and (b), the global model accuracies for $p(\mathcal{D})$ and $\tilde{p}(\mathcal{D})$ is measured on their joint distributions from 10 sampled clients.

To summarize, the knowledge on the out-local distribution $\tilde{p}(\mathcal{D})$ is prone to be forgotten in local training, resulting in the global model's forgetting. Based on our findings, we hypothesize that forgetting could be the stumbling block in federated learning.

## 2.3 Forgetting and Local Drift

First empirically observed by [56], the deviation of local updates from the desirable global direction has been widely discussed as a major cause of slow and unstable convergence in heterogeneous federated learning [21, 30, 31]. Unfortunately, given the difficulty of analyzing such drift, a common approach is to assume bounded dissimilarity between the local function gradients [20, 31].

One intriguing property of knowledge preservation on out-local distribution is that it corrects the local gradients towards the global direction. We define the gradient diversity $\Lambda$ to measure the dissimilarity of local gradients and state the effect of knowledge preservation as follows:

**Definition 2.** *For the uniformly weighted $K$ clients, the gradient diversity $\Lambda$ of local functions $f^k$ towards the global function $f = \frac{1}{K} \sum_{k=1}^{K} f^k$ is defined as:*

$$\Lambda := \frac{\frac{1}{K} \sum_{k=1}^{K} \|\nabla f^k\|^2}{\|\nabla f\|^2} \tag{8}$$

Here, $\Lambda \geqslant 1$ measures the alignment of gradient direction of the local function $f^k$s w.r.t. the global function $f$. Note that the $\Lambda$ becomes smaller as the directions of local function gradients $\nabla f^k$s become similar—e.g., if the magnitudes of the $\|\nabla f^k\|^2$s are fixed, the smallest $\Lambda$ is obtained when the direction of $\nabla f^k$s are identical. To understand the effect of preserving knowledge on the out-local distribution $\tilde{p}(\mathcal{D})$, we analyze how the local gradients and their diversity varies by adding gradient signal on $\tilde{p}(\mathcal{D})$ with factor $\beta$ and obtain the following proposition.

**Proposition 1.** *Suppose uniformly weighted $K$ clients with in-local distribution $p^k = [p_1^k, \ldots, p_C^k]$. If we assume the class-wise gradients $g_c$ are orthogonal with uniform magnitude, increasing $\beta \leqslant C/2 - 1$ reduces the gradient diversity $\Lambda$ from the local gradient $\nabla f^k = (p^k + \beta \tilde{p}^k) \cdot g$ with the ratio:*

$$\frac{\partial \Lambda}{\partial \beta} \leqslant -\frac{M_{K,C,p}}{(1+\beta)^2}. \tag{9}$$

*where $\beta$ stands for the effect of knowledge preservation on the out-local distribution $\tilde{p}^k$. The $M_{K,C,p} > 0$ is a constant term consists of $K$, $C$, and $(p^k)_{k=1}^{K}$,*

The proof is given in Appendix P. Note that here we treat $f^k$ as a sum of class-wise losses $\sum_c p_c^k \mathcal{L}_c$, where $\mathcal{L}_c = \mathbb{E}_{x|y=c}[\mathcal{L}(x; w)]$ is the loss on the specific class $c$. When $\beta = 0$, there is no regularization to preserve out-distribution knowledge, so the local model only needs to fit on the in-local distribution $p^k$. The above proposition suggests that the preserved knowledge on out-local distribution $\tilde{p}^k$ (i.e., as $\beta$ increases) guides the local gradient directions to be more aligned towards the global gradient, reducing the gradient diversity $\Lambda$. Such forgetting perspective provides the opportunity to handle the data heterogeneity at the model's prediction level.

## 3 FedNTD: Federated Not-True Distillation

In this section, we propose Federated Not-True Distillation (FedNTD) and its key features. The core idea of FedNTD is to preserve the global view only for the not-true classes. More specifically, FedNTD conducts local-side distillation by the linearly combined loss function $\mathcal{L}$ between the cross-entropy loss $\mathcal{L}_{\text{CE}}$ and the not-true distillation loss $\mathcal{L}_{\text{NTD}}$:

Figure 5: An overview of Not-True Distillation. The true class (*Class 3*) logits is ignored in the softmax.

$$\mathcal{L} = \mathcal{L}_{\text{CE}}(q^l, \mathbb{1}_y) + \beta \cdot \mathcal{L}_{\text{NTD}}(\tilde{q}_\tau^l, \tilde{q}_\tau^g). \tag{10}$$

Here, the hyperparameter $\beta$ stands for the strength of knowledge preservation on the out-local distribution. Then, the not-true distillation loss $\mathcal{L}_{\text{NTD}}$ is defined as the KL-Divergence loss between the not-true softmax prediction vector $\tilde{q}_\tau^l$ and $\tilde{q}_\tau^g$ as follows:

$$\mathcal{L}_{\text{NTD}}(\tilde{q}_\tau^l, \tilde{q}_\tau^g) = -\sum_{c=1, \mathbf{c} \neq \mathbf{y}}^{C} \tilde{q}_\tau^g(c) \log \left[ \frac{\tilde{q}_\tau^l(c)}{\tilde{q}_\tau^g(c)} \right], \quad \text{where} \quad \begin{cases} \tilde{q}_\tau^l(c) = \frac{\exp(z_c^l/\tau)}{\sum_{\tilde{\mathbf{c}} \neq \mathbf{y}}^{C} \exp(z_{\tilde{c}}^l/\tau)} \\ \\ \tilde{q}_\tau^g(c) = \frac{\exp(z_c^g/\tau)}{\sum_{\tilde{\mathbf{c}} \neq \mathbf{y}}^{C} \exp(z_{\tilde{c}}^g/\tau)} \end{cases} \quad (\forall c \neq y). \tag{11}$$

which take softmax with temperature $\tau$ only for the not-true class logits. Figure 5 illustrates how the not-true distillation works given a sample $x$. Note that ignoring the true-class logits makes gradient signal of $\mathcal{L}_{\text{NTD}}$ to the true-class as 0. The detailed algorithm is provided in Algorithm 1.

---

**Algorithm 1** Federated Not-True Distillation (FedNTD)

---

**Input:** total rounds $T$, local epochs $E$, dataset $\mathcal{D}$, sampled clients sets $K^{(t)} \subset K$ in round $t$, learning rate $\gamma$

**Initialize** $w^{(}0)$ for global server weight
**for** each communication round $t = 1, \cdots, T$ **do**
    Server samples clients $K^{(t)}$ and broadcasts $\tilde{w}^{(t)} \leftarrow w^{(t)}$
    **for** each client $k \in K^{(t)}$ **in parallel do**
        **for** Local Steps $e = 1 \cdots E$ **do**
            **for** Batches $j = 1 \cdots B$ **do**
                $\tilde{w}_k^{(t)} \leftarrow \tilde{w}_k^{(t)} - \gamma \nabla_w \mathcal{L}(\tilde{w}_k^{(t)} ; [\mathcal{D}^k]_j)$            Using [Equation 10]
        **end for**
        **end for**
    **end for**
    Upload $\tilde{w}_k^t$ to server
    **Server Aggregation :** $w^{(t+1)} \leftarrow \frac{1}{|K^{(t)}|} \sum_{k \in K^{(t)}} \tilde{w}_k^{(t)}$
**end for**
**Server output :** $w_T$

---

We now explain how learning to minimize $\mathcal{L}_{\text{NTD}}$ preserves global knowledge on out-local distribution $\tilde{p}(\mathcal{D})$. Suppose there are $N$ number of data points in the local dataset $\mathcal{D}$. The accumulated Kullback-Leibler divergence loss $\mathcal{L}_{\text{KL}}$ between $q_\tau^{l,i}$, the probability vector for the data $x_i$, and its reference $q_\tau^{g,i}$ to be matched for is:

$$\mathcal{L}_{\text{KL}} = -\frac{1}{N} \sum_{i=1}^{N} \sum_{c=1}^{\mathcal{C}} q_\tau^{g,i}(c) \log \left[ \frac{q_\tau^{l,i}(c)}{q_\tau^{g,i}(c)} \right] . \tag{12}$$

By splitting the summands for the *true* and *not-true* classes, the term becomes:

$$\mathcal{L}_{\text{KL}}^{\text{true}} = -\frac{1}{N} \sum_{i=1}^{N} q_\tau^{g,i}(y_i) \log \left[ \frac{q_\tau^{l,i}(y_i)}{q_\tau^{g,i}(y_i)} \right] , \ \mathcal{L}_{\text{KL}}^{\text{not-true}} = -\frac{1}{N} \sum_{i=1}^{N} \sum_{c' \neq y_i}^{\mathcal{C}} q_\tau^{g,i}(c') \log \left[ \frac{q_\tau^{l,i}(c')}{q_\tau^{g,i}(c')} \right] . \tag{13}$$

**Proposition 2.** *Consider the in-local distribution $p(\mathcal{D}) = [p_1 \ldots p_{\mathcal{C}}]$ such that $p_c = \frac{|\mathcal{S}_c|}{N}$ and its out-local distribution $\tilde{p}(\mathcal{D}) = [\tilde{p}_1, \ldots \tilde{p}_{\mathcal{C}}]$, where $\mathcal{S}_c$ is the set of indices satisfying $y_i = c$. Then the $\mathcal{L}_{\text{KL}}^{\text{true}}$ and $\mathcal{L}_{\text{KL}}^{\text{not-true}}$ each are equivalent to the weighted sum on $p(\mathcal{D})$ and $\tilde{p}(\mathcal{D})$ as*

$$\mathcal{L}_{\text{KL}}^{\text{true}} = \sum_{c=1}^{\mathcal{C}} \boldsymbol{p_c} \, \mathbb{E}_{i \in \mathcal{S}_c} \left[ -q_\tau^{g,i}(c) \log \left[ \frac{q_\tau^{l,i}(c)}{q_\tau^{g,i}(c)} \right] \right]$$

$$\frac{\mathcal{L}_{\text{KL}}^{\text{not-true}}}{\mathcal{C} - 1} = \sum_{c=1}^{\mathcal{C}} \boldsymbol{\tilde{p}_c} \, \mathbb{E}_{i \notin \mathcal{S}_c} \left[ -q_\tau^{g,i}(c) \log \left[ \frac{q_\tau^{l,i}(c)}{q_\tau^{g,i}(c)} \right] \right] \tag{14}$$

With a minor amount of calculation from Equation 13, we derive the above proposition. The derivation is provided in Appendix Q. The proposition suggests that matching the true-class and the not-true class logits collapses to the loss on the in-local distribution $p(\mathcal{D})$ and the out-local distribution $\tilde{p}(\mathcal{D})$.

In the loss function of our FedNTD (Equation 10), we attain the new knowledge on the in-local distribution by following the true-class signals from the labeled data in local datasets using the $\mathcal{L}_{\text{CE}}$. In the meanwhile, we preserve the previous knowledge on the out-local distribution by following the global model's perspective, corresponding to the not-true class signals, using the $\mathcal{L}_{\text{NTD}}$. Here, the hyperparameter $\beta$ controls the trade-off between learning on the new knowledge and preserving previous knowledge. This resembles to the *stability-plasticity dilemma* [39] in continual learning, where the learning methods must balance retaining knowledge from previous tasks while learning new knowledge for the current task [35].

Table 1: Accuracy@1 (%) on MNIST [11], CIFAR-10 [25], CIFAR-100 [25], and CINIC-10 [10]. The values in the parenthesis are forgetting measure $\mathcal{F}$. The arrow ($\downarrow$, $\uparrow$) shows the comparison to the FedAvg. The standard deviation of each experiment is provided in Appendix F.

| | | **NIID Partition Strategy : Sharding** | | | | | |
|---|---|---|---|---|---|---|---|
| **Method** | **MNIST** | **CIFAR-10** | | | | **CIFAR-100** | **CINIC-10** |
| | | $s = 2$ | $s = 3$ | $s = 5$ | $s = 10$ | | |
| FedAvg [37] | $78.63_{(0.20)}$ | $40.14_{(0.59)}$ | $51.10_{(0.46)}$ | $57.17_{(0.37)}$ | $64.91_{(0.26)}$ | $25.57_{(0.49)}$ | $39.64_{(0.59)}$ |
| FedCurv [43] | $78.56_{(0.21)}\downarrow$ | $44.52_{(0.53)}\uparrow$ | $49.00_{(0.47)}\downarrow$ | $54.61_{(0.39)}\downarrow$ | $62.19_{(0.27)}\downarrow$ | $22.89_{(0.49)}\downarrow$ | $40.45_{(0.57)}\uparrow$ |
| FedProx [30] | $78.26_{(0.21)}\downarrow$ | $41.48_{(0.57)}\uparrow$ | $51.65_{(0.45)}\uparrow$ | $56.88_{(0.37)}\downarrow$ | $64.65_{(0.25)}\downarrow$ | $25.10_{(0.49)}\downarrow$ | $41.47_{(0.57)}\uparrow$ |
| FedNova [47] | $77.04_{(0.21)}\downarrow$ | $42.62_{(0.56)}\uparrow$ | $52.03_{(0.44)}\uparrow$ | $62.14_{(0.30)}\uparrow$ | $66.97_{(0.20)}\uparrow$ | $26.96_{(0.41)}\uparrow$ | $42.55_{(0.56)}\uparrow$ |
| SCAFFOLD [20] | $81.05_{(0.17)}\uparrow$ | $44.60_{(0.53)}\uparrow$ | $54.26_{(0.39)}\uparrow$ | $\mathbf{65.74}_{(0.23)}\uparrow$ | $\mathbf{68.97}_{(0.16)}\uparrow$ | $30.82_{(0.36)}\uparrow$ | $42.66_{(0.54)}\uparrow$ |
| MOON [28] | $76.56_{(0.23)}\downarrow$ | $38.51_{(0.60)}\downarrow$ | $50.47_{(0.47)}\downarrow$ | $56.69_{(0.39)}\downarrow$ | $65.30_{(0.25)}\uparrow$ | $25.29_{(0.48)}\downarrow$ | $37.07_{(0.62)}\downarrow$ |
| **FedNTD (Ours)** | $\mathbf{84.44}_{(0.13)}\uparrow$ | $\mathbf{52.61}_{(0.43)}\uparrow$ | $\mathbf{58.18}_{(0.34)}\uparrow$ | $64.93_{(0.23)}\uparrow$ | $68.56_{(0.15)}\uparrow$ | $\mathbf{31.69}_{(0.32)}\uparrow$ | $\mathbf{48.07}_{(0.48)}\uparrow$ |
| | | **NIID Partition Strategy : LDA** | | | | | |
| **Method** | **MNIST** | **CIFAR-10** | | | | **CIFAR-100** | **CINIC-10** |
| | | $\alpha = 0.05$ | $\alpha = 0.1$ | $\alpha = 0.3$ | $\alpha = 0.5$ | | |
| FedAvg [37] | $79.73_{(0.19)}$ | $28.24_{(0.71)}$ | $46.49_{(0.51)}$ | $57.24_{(0.36)}$ | $62.53_{(0.28)}$ | $30.69_{(0.32)}$ | $38.14_{(0.60)}$ |
| FedCurv [43] | $78.72_{(0.20)}\downarrow$ | $33.64_{(0.66)}\uparrow$ | $44.26_{(0.53)}\downarrow$ | $54.93_{(0.38)}\downarrow$ | $59.37_{(0.30)}\downarrow$ | $29.16_{(0.32)}\downarrow$ | $36.69_{(0.61)}\downarrow$ |
| FedProx [30] | $79.25_{(0.19)}\downarrow$ | $37.19_{(0.62)}\uparrow$ | $47.65_{(0.49)}\uparrow$ | $57.35_{(0.35)}\uparrow$ | $62.39_{(0.27)}\downarrow$ | $30.60_{(0.32)}\downarrow$ | $39.47_{(0.58)}\uparrow$ |
| FedNova [47] | $60.37_{(0.38)}\downarrow$ | $10.00_{(Failed)}\downarrow$ | $28.06_{(0.71)}\downarrow$ | $57.44_{(0.35)}\uparrow$ | $64.65_{(0.23)}\uparrow$ | $32.15_{(0.28)}\uparrow$ | $30.44_{(0.68)}\downarrow$ |
| SCAFFOLD [20] | $71.57_{(0.26)}\downarrow$ | $10.00_{(Failed)}\downarrow$ | $23.12_{(0.74)}\downarrow$ | $\mathbf{62.01}_{(0.29)}\uparrow$ | $\mathbf{66.16}_{(0.19)}\uparrow$ | $\mathbf{33.68}_{(0.25)}\uparrow$ | $28.78_{(0.69)}\downarrow$ |
| MOON [28] | $78.95_{(0.20)}\downarrow$ | $28.35_{(0.71)}\uparrow$ | $44.77_{(0.53)}\downarrow$ | $58.38_{(0.35)}\uparrow$ | $63.10_{(0.27)}\uparrow$ | $30.64_{(0.32)}\downarrow$ | $37.92_{(0.61)}\downarrow$ |
| **FedNTD (Ours)** | $\mathbf{81.34}_{(0.17)}\uparrow$ | $\mathbf{40.17}_{(0.58)}\uparrow$ | $\mathbf{54.42}_{(0.42)}\uparrow$ | $62.42_{(0.29)}\uparrow$ | $66.12_{(0.19)}\uparrow$ | $32.37_{(0.26)}\uparrow$ | $\mathbf{46.24}_{(0.50)}\uparrow$ |

## 4 Experiment

### 4.1 Experimental Setup

We test our algorithm on MNIST [11], CIFAR-10 [25], CIFAR-100 [25], and CINIC-10 [10]. We distribute the data to 100 clients and randomly sample clients with a ratio of 0.1. For CINIC-10, we use 200 clients, with a sampling ratio of 0.05. We use a momentum SGD with an initial learning rate of 0.1, and the momentum is set as 0.9. The learning rate is decayed with a factor of 0.99 at each round, and a weight decay of 1e-5 is applied. We adopt two different NIID partition strategies:

- **(i) Sharding** [37]: sort the data by label and divide the data into same-sized shards, and control the heterogeneity by $s$, the number of shards per user. In this strategy only considers the statistical heterogeneity as the dataset size is identical for each client. We set $s$ as MNIST ($s = 2$), CIFAR-10 ($s \in \{2, 3, 5, 10\}$), CIFAR-100 ($s = 10$), and CINIC-10 ($s = 2$).

- **(ii) Latent Dirichlet Allocation (LDA)** [34, 46]: assigns partition of class $c$ by sampling $p_c \approx \mathrm{Dir}(\alpha)$. In this strategy, both the distribution and dataset size are different for each client. We set $\alpha$ as MNIST ($\alpha = 0.1$), CIFAR-10 ($\alpha \in \{0.05, 0.1, 0.3, 0.5\}$), CIFAR-100 ($\alpha = 0.1$), and CINIC-10 ($\alpha = 0.1$)

More details on model, datasets, hyperparameters, and partition strategies are provided in Appendix B.

### 4.2 Performance on Data Heterogeneity

We compare our FedNTD with various existing works, with results shown in Table 1. As reported in [27], even the state-of-the-art methods perform well only in specific setups, and their performance often deteriorates below FedAvg. However, our FedNTD consistently outperforms the baselines on all setups, achieving state-of-the-art results in most cases.

For each experiment in Table 1, we also report the forgetting $\mathcal{F}$ in the parenthesis along with the accuracy. Note that the smaller $\mathcal{F}$ value indicates the global model less forgets the previous knowledge. We find that the performance in federated learning is closely related to forgetting, improving as the forgetting reduces. *We believe that the gain from the prior works is actually from the forgetting prevention in their own ways.*

We emphasize that the prior works to learn from heterogeneous locals often require *statefulness* (i.e., clients should be repeatedly sampled with identification) [28, 47], *additional communication cost* [20], or *auxiliary data* [33]. However, our FedNTD neither compromise any potential privacy issue nor requires additional communication burden. A brief comparison is provided in Appendix C.

We further conduct experiments on the effect of local epochs, client sampling ratio, and model architecture is in Appendix G, the advantage of not-true distillation over knowledge distillation in Appendix H, and the effect of hyperparameters of FedNTD in Appendix K. In the next section, we analyze how the knowledge preservation of FedNTD benefits on the federated learning.

## 5 Knowledge preservation of FedNTD

In Figure 7, we present the test accuracy on diverse heterogeneity levels. Although both FedAvg and FedNTD show little change in local accuracy on in-local distribution $p(x)$, FedNTD significantly improves the local accuracy on out-local distribution $\tilde{p}(x)$, which implies it prevents forgetting. Along with it, the test accuracy of the global model also substantially improves. These gaps are enlarged when the number of local epochs increases, where the local models much deviate from the global model. The accuracy curves during local training are in Figure 6. It shows that fitting on the local distribution rapidly

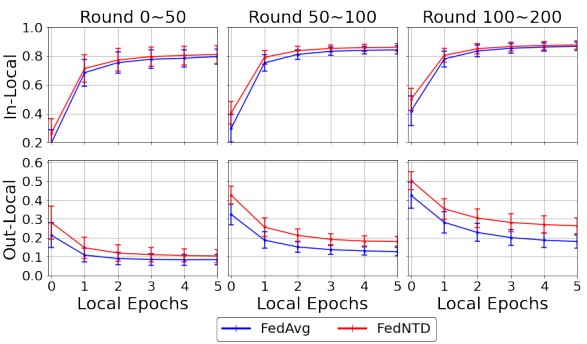

Figure 6: Accuracy on CIFAR-10 (s=2) in *Local Training*. The error bars stand for the standard deviation on clients.

leads to forgetting on out-local distribution, But FedNTD effectively relieves this tendency without hurting the learning ability towards the in-local distribution.

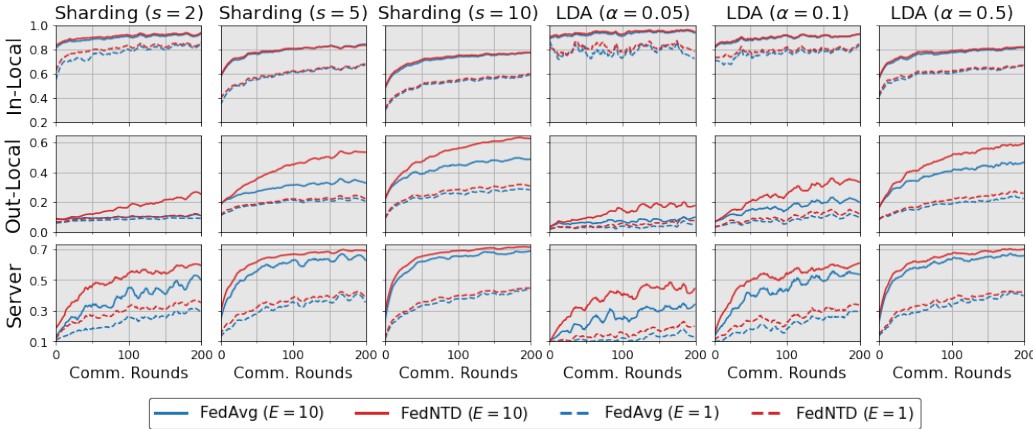

Figure 7: Learning curves of FedAvg [37] (*blue line*) and our FedNTD (*red line*) on CIFAR-10 with various heterogeneity setups for local epochs $E \in \{1, 10\}$. (1st and 2nd row) : Local test accuracy on in-local distribution $p(x)$ and out-local distribution $\tilde{p}(x)$. (3rd row): Global server test accuracy.

Our interest is how the FedNTD's knowledge preservation on out-local distribution benefits on the federated learning, despite little change of its performance on in-local distribution. To figure it out, we analyze the local models in FedNTD after local training, and suggest two main reasons:

- **Weight Alignment**: *How much the semantics of each weight is preserved?*
- **Weight Divergence**: *How far the local weights drift from the global model?*

### 5.1 Weight Alignment

In recent studies, it has been suggested that there is a mismatch of semantic information encoded in the weight parameters across local models, even for the same coordinates (i.e., same position)

[46, 53, 54]. As the current aggregation scheme averages weights of the identical coordinates, matching the semantic alignment across local models plays an important role in global convergence.

To analyze the semantic alignment of each parameter, we identify the individual neuron's class preference by which class has the largest activation output on average. We then measure the alignment for a layer between two different models as the proportion of neurons that the class preference is matched for each other. The result is provided in Table 2.

While FedAvg and FedNTD show little difference in the IID case, FedNTD significantly enhances the alignment in the NIID cases. The visualized results are in Appendix M, with more details on how the alignment is measured. We further analyze the change of feature after local training using a unit hypersphere in Appendix N and T-SNE in Appendix O.

Table 2: Alignment of last two fc-layers for Distributed Global (DG), 10 Locals (L), and Aggregated Global (AG) models on CIFAR-10 datasets for IID and NIID (Sharding $s = 2$, LDA $\alpha = 0.05$) at round 200.

| Layer | Alignment | IID | | NIID ($s = 2$) | | NIID ($\alpha = 0.05$) | |
|---|---|---|---|---|---|---|---|
| | | FedAvg | FedNTD | FedAvg | FedNTD | FedAvg | FedNTD |
| Linear_1 | $W_G^{(t-1)}$ vs. $W_L^{(t)}$ | 0.679 | 0.668 | 0.635 | **0.703** | 0.597 | **0.756** |
| (dim: 512) | $W_G^{(t-1)}$ vs. $W_G^{(t)}$ | 0.850 | 0.830 | 0.787 | **0.871** | 0.670 | **0.856** |
| Linear_1 | $W_G^{(t-1)}$ vs. $W_L^{(t)}$ | 0.771 | 0.765 | 0.488 | **0.552** | 0.512 | **0.730** |
| (dim: 128) | $W_G^{(t-1)}$ vs. $W_G^{(t)}$ | 0.898 | 0.906 | 0.609 | **0.836** | 0.586 | **0.859** |

## 5.2 Weight Divergence

The knowledge preservation by FedNTD leads the global model to predict each class more evenly. Here we describe how the global model with even prediction performance stabilizes the weight divergence. Consider a model fitted on a specific original distribution, and now it is trained on a new distribution. Then the weight distance between the original model and fitted model increases as the distance between the original distribution and new distribution grows.

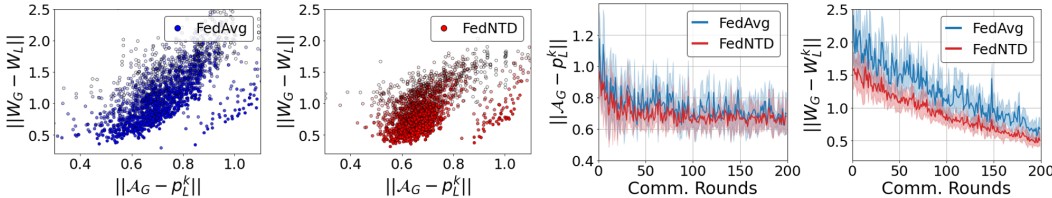

Figure 8: Distances for weights and distributions on CIFAR10 (s=2). (a), (b): The relationship between two distances. The opacity is higher for later rounds. (c), (d): The measured distances for 200 rounds.

We argue that if the distance between the global model's underlying distribution and local distributions is small, the moved distance between the global model and local models also becomes close. If we assume the local distributions are generated arbitrarily, the most robust choice for the global model's underlying distribution is a uniform distribution. We formally rephrase our argument as the follows:

**Proposition 3.** *Let* $\mathbb{P} : \Delta_{\mathcal{C}} \to \mathbb{R}_{\geqslant 0}$ *be the probability measure for the client's local distribution and* $\Pi$ *be the set of measure in the hypothesis. Assume that the class-wise loss* $\mathcal{L}_c(w) = \mathbb{E}_{x|y=c}[\mathcal{L}(x; w)]$ *is* $\lambda$-*smooth and* $w_c$ *be a optimum of* $\mathcal{L}_c$. *Then the loss* $\mathcal{L}$ *of the client with distribution* $p$ *becomes:*

$$\mathcal{L}(w) = \sum_c p_c \mathcal{L}_c(w) \leqslant \sum_c p_c \mathcal{L}_c(w_c) + \frac{1}{2}\lambda \sum_c p_c \|w - w_c\|^2 \,. \tag{15}$$

*Then for arbitrary* $\mathbb{P} \in \Pi$, *the uniform distribution attains the minimax value:*

$$unif.dist \in \underset{p \in \Delta_{\mathcal{C}}}{\operatorname{argmin}} \sup_{\mathbb{P} \in \Pi} \mathbb{E}_{p' \sim \mathbb{P}}[\|w_{p'} - w_p\|] , \quad \text{where } w_p = \sum_c p_c w_c \,. \tag{16}$$

The proof is provided in Appendix S. Although the global model's underlying distribution is unknown, the normalized class-wise accuracy vector is a handy approximation for it as: $\mathcal{A}_{\text{G}} = \frac{1}{A} \cdot [a_1, \dots a_{\mathcal{C}}]$, where $A$ is the global model's test accuracy and $a_c$ is its class-wise accuracy on the class $c$.

The results in Figure 8 empirically validate our argument. There is a strong correlation between weight divergence $\|w - w_k\|$ (for global model $w$ and client $k$'s local model $w_k$) and distribution distance $\|\mathcal{A}_G - p_k\|$ (for client $k$'s distribution $p_k$). By providing a better starting point for local training, FedNTD effectively stabilizes the weight divergence.

## 6 Related Work

**Federated Learning (FL)** is proposed to update a global model while the local data is kept in clients' devices [23, 24]. The standard algorithm is FedAvg [37], which aggregates trained local models by averaging their parameters. Although its effectiveness has been largely discussed in i.i.d. settings [44, 48], many algorithms obtain the sub-optimal when the distributed data is heterogeneous [31, 56]. Until recently, a wide range of variants of FedAvg has been proposed to overcome such a problem. One line of work focuses on *local-side* modification by regularizing the deviation of local models from the global model [1, 20, 30, 47]. Another is the *server-side* modification, which improves the efficacy of aggregation of local models in the server [9, 33, 46, 55]. Our work aims to preserve global knowledge during local training, which belongs to the local-side approach.

**Forgetting View in FL** A pioneer work that considers forgetting in FL is FedCurv [43]. It regards each local client as a task, and [43] regulates the change of local parameters to prevent accuracy drop on all other clients. However, it needs to compute and communicate parameter-wise importance across clients, which severely burdens the learning process. On the other hand, we focus on the class-wise forgetting and suggest that not-true logits from local data contain enough knowledge to prevent it. A concurrent work of our study is [49], which also reports the forgetting issue in local clients by empirically showing the increasing loss of previously learned data after the local training. To prevent forgetting, [49] exploits generated pseudo data. Instead, we focus on the class-wise forgetting and suggest that not-true logits from local data contain enough knowledge to prevent it. The continual learning literature is further discussed in Appendix D.

**Knowledge Distillation (KD) in FL** In FL, a typical approach is using KD to make the global model learn from the ensemble of local models [9, 26, 33, 57]. By leveraging the unlabeled auxiliary data, KD effectively tackles the local drifts by enriching the aggregation. However, such carefully engineered proxy data and may not always be available [30, 55, 58]. Although more recent works generate pseudo-data to extract knowledge by data-free KD [55, 58], they require additional heavy computation, and the quality of samples is sensitive to the many hyperparameters involved in the process. On the other hand, as a simple variant of KD, our proposed method surprisingly performs well on heterogeneity scenarios without any additional resource requirements.

## 7 Conclusion

This study begins from an analogy to continual learning and suggests that forgetting could be a major concern in federated learning. Our observations show that the knowledge outside of local distribution is prone to be forgotten in local training and is closely related to the unstable global convergence. To overcome this issue, we propose a simple yet effective algorithm, FedNTD, which conducts local-side distillation only for the not-true classes to prevent forgetting. FedNTD does not have any additional requirements, unlike previous approaches. We analyze the effect of FedNTD from various perspectives and demonstrate its benefits in federated learning.

**Broader Impact** We believe that Federated Learning is an important learning paradigm that enables privacy-preserving ML. Our work suggests the forgetting issue and introduces the methods to relive it without compromising data privacy. The insight behind this work may inspire new researches. However, the proposed method maintains the knowledge outside of local distribution in the global model. This implies that if the global model is biased, the trained local model is more prone to have a similar tendency. This should be considered for ML participators.

## Acknowledgments

This work was supported by Institute of Information & communications Technology Planning & Evaluation (IITP) grant funded by Korea government (MSIT) [No. 2021-0-00907, Development of Adaptive and Lightweight Edge-Collaborative Analysis Technology for Enabling Proactively Immediate Response and Rapid Learning, 90%] and [No. 2019-0-00075, Artificial Intelligence Graduate School Program (KAIST), 10%].

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
