# A  Table of Notations

Table 3: Table of Notations throughout the paper.

| | |
|---|---|
| **Indices:** | |
| $c$, $c'$ | index for classes ($c \in \{1, ..., \mathcal{C}\} = [\mathcal{C}]$) |
| $i$ | index for data ($i \in \{1, ..., N\} = [N]$) |
| $k$, $k'$ | index for clients ($k \in \{1, ..., K\} = [K]$ or $\in K^{(t)}$) |
| $t$ | index for rounds ($t \in \{1, ..., \mathcal{T}\} = [\mathcal{T}]$) |
| $e$ | index for local epochs ($t \in \{1, ..., E\} = [E]$) |
| **Parameters:** | |
| $\alpha$ | Parameter for the Dirichlet Distribution |
| $s$ | The Number of shards per user |
| $\beta$ | Hyperparameter for the distillation loss; generally controls the relative weight of divergence loss |
| $\tau$ | The temperature on the softmax |
| $\gamma$ | Learning rate |
| **Data and Weights:** | |
| $\mathcal{D}$ | whole dataset |
| $\mathcal{D}^k$ | local dataset |
| $x$ | datum |
| $y$ | class label for datum |
| $w^{(t)}$ | weight of the server model on the round $t$ |
| $w_k^{(t)}$ | weight of the $k$-th client model on the round $t$ |
| $\|W_G - W_L\|$ | Collection of $L^1$-norm between server and client models, among all rounds. |
| **Softmax Probabilities:** | |
| $q_\tau^T$ / $q_\tau^S$ | The softened softmax probability of teacher/student model |
| $q^g$ / $q^l$ | The softmax probability on the server/client model |
| $\tilde{q}_\tau^g$ / $\tilde{q}_\tau^l$ | The softened softmax probability calculated without true-class logit on the server/client model |
| **Class Distribution on Datasets:** | |
| $p = [p_1, \ldots, p_{\mathcal{C}}]$ | In-local distribution of dataset |
| $\tilde{p} = [\tilde{p}_1, \ldots, \tilde{p}_{\mathcal{C}}]$ | Out-local distribution of dataset |
| **Loss Functions:** | |
| $\mathcal{L}_{\text{CE}}$ | cross-entropy loss |
| $\mathcal{L}_{\text{KL}}$ | Kullback-Leibler divergence |
| $\mathcal{L}_{\text{NTD}}$ | Proposed Not-True Distillation Loss |
| **Accuracy and Forgetting Measure:** | |
| $\mathcal{A}_c^{(t)}$ | Accuracy of the server model on the $c$-th class, at round $t$. |
| $\|\mathcal{A}_G - p_L^k\|$ | Collection of $L^1$-norm between *normalized* global accuracy and data distribution on each client, among all rounds. |
| $\mathcal{F}$ | Backward Transfer (BwT). For the federated learning situation, we calculate this measure on the server model. |

# B Experimental Setups

Here we provide details of our experimental setups. The code is implemented by PyTorch [41] and the overall code structure is based on FedML [16] library with some modifications. We use 1 Titan-RTX and 1 RTX 2080Ti GPU card. Multi-GPU training is not conducted in the paper experiments.

## B.1 Model Architecture

The model architecture used in our experiment is from [37], which is composed of two convolutional layers followed by max-pooling layers, and two fully connected layer. A similar architecture is used in [28, 34].

## B.2 Datasets

We mainly used four benchmarks: MNIST [11], CIFAR-10 [25], CIFAR-100 [25], and CINIC-10 [10]. The details about each datasets and setups are described in Table 4. We augment the training data using Random Cropping, Horizontal Flipping, and Normalization. For MNIST, CIFAR-10, and CIFAR-100, we add Cutout [12] augmentation.

Table 4: Details datasets setups used in the experiment.

| Datasets | MNIST | CIFAR-10 | CIFAR-100 | CINIC-10 |
|---|---|---|---|---|
| Datasets Classes | 10 | 10 | 100 | 10 |
| Datasets Size | 50,000 | 50,000 | 50,000 | 90,000 |
| Number of Clients | 100 | 100 | 100 | 200 |
| Client Sampling Ratio | 0.1 | 0.1 | 0.1 | 0.05 |
| Local Epochs (E) | 3 | 5 | 5 | 5 |
| Batch Size (B) | 50 | 50 | 50 | 50 |

## B.3 Learning Setups

We use a momentum SGD optimizer with an initial learning rate of 0.01, and the momentum is set as 0.9. The momentum is only used in the local training, which implies the momentum information is not communicated to the server. The learning rate is decayed with a factor of 0.99 at each round, and a weight decay of 0.00001 is applied. In the motivational experiment in Section 3, we fix the learning rate as 0.01. Since we assume a synchronized federated learning scenario, parallel distributed learning is simulated by sequentially training the sampled clients and then aggregating them as a global model.

## B.4 Algorithm Implementation Details

For the implemented algorithms, we search hyperparameters and choose the best among the candidates. The hyperparameters for each algorithm is in Table 5.

Table 5: Algorithm-specific hyperparameters used in the experiment.

| Method | Hyperparameters | Searched Candidates |
|---|---|---|
| FedAvg [37] | None | None |
| FedCurv [43] | $s = 500, \lambda = 1.0$ | $s \in \{250, 500\}, \lambda \in \{0.1, 0.5, 1.0\}$ |
| FedProx [30] | $\mu = 0.1$ | $\mu \in \{0.1, 0.5, 1.0\}$ |
| FedNova [47] | None | None |
| SCAFFOLD [20] | None | None |
| MOON [28] | $\mu = 1.0, \tau = 0.5$ | $\mu \in \{0.1, 0.5, 1.0\}, \tau \in \{0.1, 0.5, 1, 0\}$ |
| **FedNTD (Ours)** | $\beta{=}1.0, \tau = 1.0$ | None |

## B.5 Non-IID Partition Strategy

To widely address the heterogeneous federated learning scenarios, we distribute the data to the local clients with the following two strategies: (1) **Sharding** and (2) **Latent Dirichlet Allocation** (LDA).

**Sharding** In the Sharding strategy, we sort the data by label and divide it into the same-sized shards without overlapping. In detail, a shard contains $\frac{|D|}{N \times s}$ size of same class samples, where $D$ is the total dataset size, $N$ is the total number of clients, and $s$ is the number of shards per user. Then, we distribute $s$ number of shards to each client. $s$ controls the heterogeneity of local data distribution. The heterogeneity level increases as the shard per user $s$ becomes smaller and vice versa. Note that we only test the statistical heterogeneity (skewness of class distribution across the local clients) in the Sharding strategy, and the size of local datasets is identical.

**Latent Dirichlet Allocation (LDA)** In LDA strategy, Each client $k$ is allocated with $p_{c,k}$ proportion of the training samples of class $c$, where $p_c \sim \text{Dir}_K(\alpha)$ and $\alpha$ is the concentration parameter controlling the heterogeneity. The heterogeneity level increases as the concentration parameter $\alpha$ becomes smaller, and vice versa. Note that both the class distribution and local datasets sizes differ across the local clients in LDA strategy.

## C   Conceptual comparison to prior works

The conceptual illustration of federated distillation methods is in Figure 9. The existing algorithms either use additional local information (Figure 9a) or need an auxiliary (or *proxy*) data to conduct distillation. On the other hand, our proposed FedNTD does not have such constraints (Figure 9c).

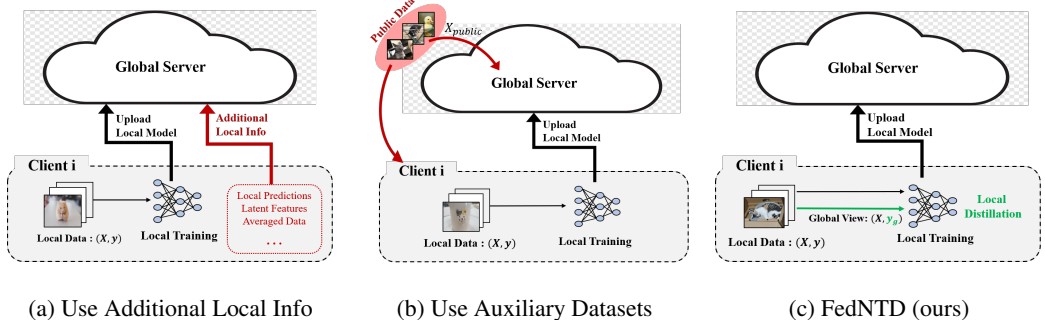

|  | (a) Use Additional Local Info | (b) Use Auxiliary Datasets | (c) FedNTD (ours) |

Figure 9: An overview of federated distillation methods.

Table 6: Additional resource requirements compared to FedAvg.

| Method | No Additional Requirements on: | | |
|---|---|---|---|
|  | Statefulness? | Communication Cost? | Auxiliary Data? |
| FedEnsemble [33] | ✔ | ✔ | ✘ |
| FedBE [9] | ✔ | ✘ | ✘ |
| MOON [28] | ✘ | ✔ | ✔ |
| SCAFFOLD [20] | ✘ | ✘ | ✔ |
| **FedNTD (Ours)** | ✔ | ✔ | ✔ |

# D  Related Work: Continual Learning

Continual Learning (CL) is a learning paradigm that updates a sequence of tasks instead of training on the whole datasets at once [42, 45]. In CL, the main challenge is to avoid catastrophic forgetting [15], whereby training on the new task interferes with the previous tasks. Existing methods try to mitigate this problem by various strategies. In *Parameter-based* approaches, the importance of parameters for the previous task is measured to restrict their changes [3, 5, 22]. *Regularization-based* approaches [4, 32] introduce regularization terms to prevent forgetting. *Memory-based* approaches [6, 8] keep a small episodic memory from the previous tasks and replay it to maintain knowledge. Our work is more related to the regularization-based approaches, introducing the additional local objective term to prevent forgetting out-local knowledge.

It would be worth to mention that there are some works that tried to conduct classical continual learning problems in federated learning setups. For example, [51] studied the scenario in which each local client has to learn a sequence of tasks. Here, the task-specific parameters are decomposed from the global parameters to minimize the interference between tasks. In [13], the relation knowledge for the old classes is transferred round by round with class-aware gradient compensations.

# E  Server Prediction Consistency

We extend the motivational experiment in Section 3.1 to the main experimental setups. Here, we plotted the *normalized* class-wise test accuracy for each case to identify the contribution of each class on the current accuracy. This helps observe the prediction consistency regardless of the highly fluctuating global server accuracy in the non-IID case. As in Figure 10, FedNTD effectively preserves the knowledge from the previous rounds; thereby the global server model becomes to predict each class evenly much earlier than FedAvg. Note that we normalize the class-wise test accuracy as round-by-round manner, which makes the sum for each round as 1.0.

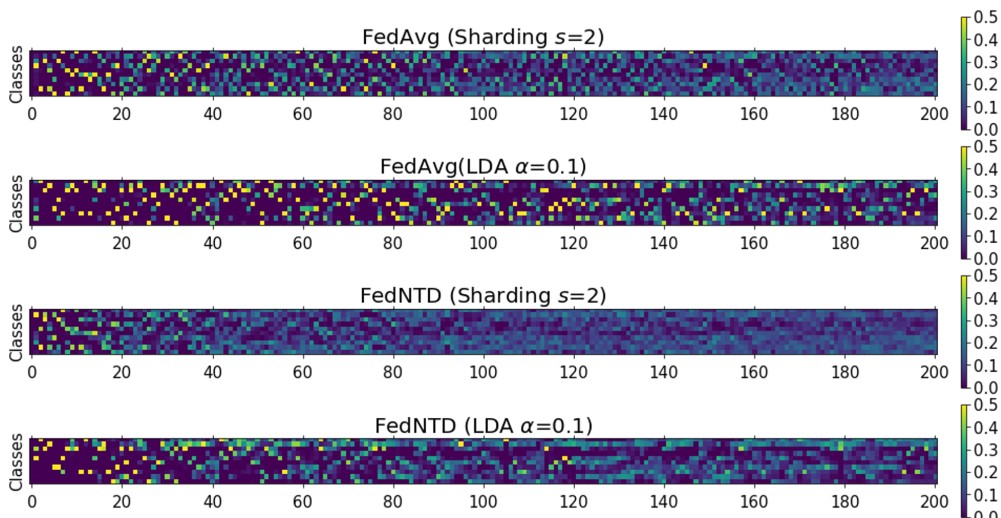

Figure 10: Visualized server test accuracy of FedAvg and FedNTD on CIFAR-10 datasets.

# F  Experiment Table with Standard Deviation

Table 7: Accuracy@1 (%) on MNIST [11], CIFAR-10 [25], CIFAR-100 [25], and CINIC-10 [10]. The values in the parenthesis are the standard deviation. The arrow (↓, ↑) shows the comparison to the FedAvg.

| | | NIID Partition Strategy : Sharding | | | | | |
|---|---|---|---|---|---|---|---|
| **Method** | **MNIST** | **CIFAR-10** | | | | **CIFAR-100** | **CINIC-10** |
| | | $s=2$ | $s=3$ | $s=5$ | $s=10$ | | |
| FedAvg [37] | $78.63_{+0.42}$ | $40.14_{+1.15}$ | $51.10_{+0.11}$ | $57.17_{+0.12}$ | $64.91_{+0.69}$ | $25.57_{+0.44}$ | $39.64_{+0.78}$ |
| FedCurv [43] | $78.56_{+0.23}$ ↓ | $44.52_{+0.44}$ ↑ | $49.00_{+0.41}$ ↓ | $54.61_{+0.20}$ ↓ | $62.19_{+0.47}$ ↓ | $22.89_{+0.66}$ ↓ | $40.45_{+0.25}$ ↑ |
| FedProx [30] | $78.26_{+0.28}$ ↓ | $41.48_{+1.08}$ ↑ | $51.65_{+0.53}$ ↑ | $56.88_{+0.15}$ ↓ | $64.65_{+0.61}$ ↓ | $25.10_{+0.67}$ ↓ | $41.47_{+0.99}$ ↑ |
| FedNova [47] | $77.04_{+0.98}$ ↓ | $42.62_{+1.32}$ ↑ | $52.03_{+1.49}$ ↑ | $62.14_{+0.74}$ ↑ | $66.97_{+0.39}$ ↑ | $26.96_{+0.59}$ ↑ | $42.55_{+0.10}$ ↑ |
| SCAFFOLD [20] | $81.05_{+0.26}$ ↑ | $44.60_{+2.24}$ ↑ | $54.26_{+0.22}$ ↑ | $\mathbf{65.74}_{+0.36}$ ↑ | $\mathbf{68.97}_{+0.34}$ ↑ | $30.82_{+0.31}$ ↑ | $42.66_{+0.92}$ ↑ |
| MOON [28] | $76.56_{+0.24}$ ↓ | $38.51_{+0.96}$ ↓ | $50.47_{+0.15}$ ↓ | $56.69_{+0.11}$ ↓ | $65.30_{+0.51}$ ↑ | $25.29_{+0.24}$ ↓ | $37.07_{+0.24}$ ↓ |
| **FedNTD (Ours)** | $\mathbf{84.44}_{+0.43}$ ↑ | $\mathbf{52.61}_{+1.00}$ ↑ | $\mathbf{58.18}_{+1.42}$ ↑ | $64.93_{+0.34}$ ↑ | $68.56_{+0.24}$ ↑ | $\mathbf{31.69}_{+0.13}$ ↑ | $\mathbf{48.07}_{+0.36}$ ↑ |

| | | NIID Partition Strategy : LDA | | | | | |
|---|---|---|---|---|---|---|---|
| **Method** | **MNIST** | **CIFAR-10** | | | | **CIFAR-100** | **CINIC-10** |
| | | $\alpha=0.05$ | $\alpha=0.1$ | $\alpha=0.3$ | $\alpha=0.5$ | | |
| FedAvg [37] | $79.73_{+0.20}$ | $28.24_{+3.11}$ | $46.49_{+0.93}$ | $57.24_{+0.21}$ | $62.53_{+0.41}$ | $30.69_{+0.27}$ | $38.14_{+3.40}$ |
| FedCurv [43] | $78.72_{+0.44}$ ↓ | $33.64_{+2.98}$ ↑ | $44.26_{+0.79}$ ↓ | $54.93_{+0.46}$ ↓ | $59.37_{+0.24}$ ↓ | $29.16_{+0.22}$ ↓ | $36.69_{+3.03}$ ↓ |
| FedProx [30] | $79.25_{+0.16}$ ↓ | $37.19_{+3.17}$ ↑ | $47.65_{+0.90}$ ↑ | $57.35_{+0.40}$ ↑ | $62.39_{+0.31}$ ↓ | $30.60_{+0.16}$ ↓ | $39.47_{+3.40}$ ↑ |
| FedNova [47] | $60.37_{+2.71}$ ↓ | $10.00$ (*Failed*) ↓ | $28.06_{+0.12}$ ↓ | $57.44_{+1.69}$ ↑ | $64.65_{+0.34}$ ↑ | $32.15_{+0.13}$ ↑ | $30.44_{+1.35}$ ↓ |
| SCAFFOLD [20] | $71.57_{+0.72}$ ↓ | $10.00$ (*Failed*) ↓ | $23.12_{+0.55}$ ↓ | $\mathbf{62.01}_{+0.34}$ ↑ | $\mathbf{66.16}_{+0.13}$ ↑ | $\mathbf{33.68}_{+0.13}$ ↑ | $28.78_{+1.26}$ ↓ |
| MOON [28] | $78.95_{+0.46}$ ↓ | $28.35_{+3.68}$ ↑ | $44.77_{+1.12}$ ↓ | $58.38_{+0.09}$ ↑ | $63.10_{+0.00}$ ↑ | $30.64_{+0.12}$ ↓ | $37.92_{+3.31}$ ↓ |
| **FedNTD (Ours)** | $\mathbf{81.34}_{+0.33}$ ↑ | $\mathbf{40.17}_{+3.19}$ ↑ | $\mathbf{54.42}_{+0.06}$ ↑ | $62.42_{+0.53}$ ↑ | $66.12_{+0.26}$ ↑ | $32.37_{+0.02}$ ↑ | $\mathbf{46.24}_{+1.67}$ ↑ |

# G  Additional Experiments

## G.1  Effect of Local Epochs

Table 8: Accuracy@1 on CIFAR-10 (Sharding $s=2$). The value in the parenthesis is the forgetting $\mathcal{F}$.

| | NIID Partition Strategy: Sharding ($s=2$) | | | | |
|---|---|---|---|---|---|
| **Method** | **Local Epochs (E)** | | | | |
| | 1 | 3 | 5 | 10 | 20 |
| FedAvg [37] | 29.49 (0.70) | 34.49 (0.64) | 40.14 (0.59) | 50.08 (0.49) | 56.93 (0.42) |
| FedProx [30] | 29.44 (0.69) ↓ | 34.00 (0.64) ↓ | 41.48 (0.57) ↑ | 42.74 (0.53) ↓ | 43.39 (0.52) ↓ |
| FedNova [47] | 27.77 (0.71) ↓ | 32.00 (0.64) ↓ | 42.62 (0.56) ↑ | 48.59 (0.50) ↓ | 58.24 (0.39) ↑ |
| SCAFFOLD [20] | 34.46 (0.64) ↑ | 39.26 (0.58) ↑ | 44.60 (0.53) ↑ | 55.35 (0.41) ↑ | **62.80** (0.34) ↑ |
| **FedNTD (Ours)** | **35.77** (0.64) ↑ | **45.47** (0.50) ↑ | **52.61** (0.43) ↑ | **60.22** (0.36) ↑ | 60.61 (0.34) ↑ |

Table 9: Accuracy@1 on CIFAR-10 (LDA $\alpha=0.1$). The value in the parenthesis is the forgetting $\mathcal{F}$.

| | NIID Partition Strategy: LDA ($\alpha=0.1$) | | | | |
|---|---|---|---|---|---|
| **Method** | **Local Epochs (E)** | | | | |
| | 1 | 3 | 5 | 10 | 20 |
| FedAvg [37] | 29.77 (0.69) | 37.70 (0.60) | 46.49 (0.51) | 53.80 (0.43) | 57.70 (0.39) |
| FedProx [30] | 33.37 (0.65) ↑ | 37.88 (0.57) ↑ | 47.65 (0.49) ↑ | 44.02 (0.50) ↓ | 44.98 (0.49) ↓ |
| FedNova [47] | 26.35 (0.73) ↓ | 24.37 (0.74) ↓ | 28.06 (0.71) ↓ | 47.41 (0.50) ↓ | 10.00 (*Failure*) ↓ |
| SCAFFOLD [20] | 13.36 (0.86) ↓ | 22.04 (0.75) ↓ | 23.12 (0.74) ↓ | 38.49 (0.57) ↓ | 47.07 (0.47) ↓ |
| **FedNTD (Ours)** | **33.94** (0.64) ↑ | **45.92** (0.50) ↑ | **54.42** (0.42) ↑ | **60.67** (0.33) ↑ | **62.25** (0.30) ↑ |

 **G.2   Effect of Sampling Ratio**

Table 10: Accuracy@1 on CIFAR-10 (Sharding $s = 2$). The value in the parenthesis is the forgetting $\mathcal{F}$.

| Method | Client Sampling Ratio (R) | | | | |
|---|---|---|---|---|---|
| | 0.05 | 0.1 | 0.3 | 0.5 | 1.0 |
| FedAvg [37] | 33.06 (0.66) | 40.14 (0.59) | 49.99 (0.46) | 52.98 (0.41) | 51.48 (0.30) |
| FedProx [30] | 35.36 (0.63) ↑ | 41.48 (0.57) ↑ | 44.54 (0.45) ↓ | 50.02 (0.31) ↓ | 52.53 (0.06) ↑ |
| FedNova [47] | 29.99 (0.69) ↓ | 42.62 (0.56) ↑ | 55.59 (0.31) ↑ | 56.75 (0.23) ↑ | 51.89 (0.34) ↑ |
| SCAFFOLD [20] | 29.15 (0.70) ↓ | 44.60 (0.53) ↑ | 55.59 (0.31) ↑ | 56.75 (0.23) ↑ | 57.88 (0.10) ↑ |
| **FedNTD (Ours)** | **46.99** (0.51) ↑ | **52.61** (0.43) ↑ | **59.37** (0.28) ↑ | **60.70** (0.18) ↑ | **61.53** (0.04) ↑ |

Table 11: Accuracy@1 on CIFAR-10 (LDA $\alpha = 0.1$). The value in the parenthesis is the forgetting $\mathcal{F}$.

| Method | Client Sampling Ratio (R) | | | | |
|---|---|---|---|---|---|
| | 0.05 | 0.1 | 0.3 | 0.5 | 1.0 |
| FedAvg [37] | 29.35 (0.70) | 46.49 (0.51) | 53.73 (0.39) | 58.72 (0.25) | 61.38 (0.04) |
| FedProx [30] | 36.36 (0.63) ↑ | 47.65 (0.49) ↑ | 45.78 (0.37) ↓ | 49.65 (0.23) ↓ | 51.31 (0.07)) ↓ |
| FedNova [47] | 21.31 (0.78) ↓ | 28.06 (0.71) ↓ | 45.83 (0.49) ↓ | 55.09 (0.50) ↓ | 56.79 (0.30) ↓ |
| SCAFFOLD [20] | 15.80 (0.84) ↓ | 23.12 (0.74) ↓ | 41.29 (0.51) ↓ | 10.00 (*Failure*) ↓ | 10.00 (*Failure*) ↓ |
| **FedNTD (Ours)** | **45.80** (0.53) ↑ | **54.42** (0.42) ↑ | **58.57** (0.33) ↑ | **60.88** (0.19) ↑ | **62.48** (0.06) ↑ |

## G.3   Results on ResNet-10 Model

We report an additional experiment on popular architecture, ResNet-10. The number of parameters in ResNet-10 is about 10x larger than the 2-conv + 2-fc model for the main experiments.

| | FedAvg | Scaffold | MOON | FedNTD (ours) |
|---|---|---|---|---|
| **Shard ($s = 2$)** | 36.01 | 44.59 | 35.21 | **46.27** |
| **Shard ($s = 5$)** | 39.21 | 65.08 | 51.02 | **65.92** |
| **LDA ($\alpha = 0.1$)** | 33.35 | 38.78 | 33.57 | **49.85** |

# H   Comparison to KD

We analyze the advantage of FedNTD over KD by observing the performance of the loss function below. Note that $L(\lambda)$ moves from $\mathcal{L}_{\text{KD}}$ to $\mathcal{L}_{\text{NTD}}$ as $\lambda$ increases, and collapses to $\mathcal{L}_{\text{KD}}$ at $\lambda = 0$ and $\mathcal{L}_{\text{NTD}}$ at and $\lambda = 1$.

$$\mathcal{L}_{\text{KD}\to\text{NTD}} = \mathcal{L}_{\text{CE}}(q, \mathbb{1}_y) + L(\lambda), \tag{17}$$

$$L(\lambda) = (1 - \lambda) \cdot \mathcal{L}_{\text{KD}}(q_\tau^l, \ q_\tau^g) + \lambda \cdot \mathcal{L}_{\text{NTD}}(\tilde{q}_\tau^l, \ \tilde{q}_\tau^g). \tag{18}$$

The result is in Table 12, which shows reaching $L(\lambda)$ to $\mathcal{L}_{NTD}$ significantly improves the performance. This improvement supports the effect of decoupling the not-true classes and the true classes: preservation of out-local distribution knowledge using not-true class signals and acquisition of new knowledge on true classes from local datasets.

Table 12: CIFAR-10 test accuracy by varying $\lambda$.

| Partition Method | KD | KD → NTD | | | | | NTD |
|---|---|---|---|---|---|---|---|
| | | 0.1 | 0.3 | 0.5 | 0.7 | 0.9 | |
| Sharding ($s = 2$) | 46.2 | 46.4 | 46.9 | 47.6 | 48.8 | 50.2 | **52.6** |
| LDA ($\alpha = 0.1$) | 50.8 | 50.9 | 51.4 | 51.9 | 52.6 | 53.6 | **54.9** |

To further analyze the effect of NTD, we measure the performance of the local model as the communication rounds proceed. The result is plotted in Figure 11. Note that the *Personalized* performance is evaluated on the test samples with the same label distribution of the local clients.

The result shows that although the KD considerably improves the server model performance, the local model learned by KD shows much lower local performance. On the other hand, FedNTD shows much higher local performance, which implies that NTD successfully tackles the distillation not to hinder the local learning.

We insist that such significant improvement by discarding true-class logits in the distillation loss comes from the better trade-off between attaining new knowledge from local data and preserving old knowledge in the global model, as suggested in Section 3.

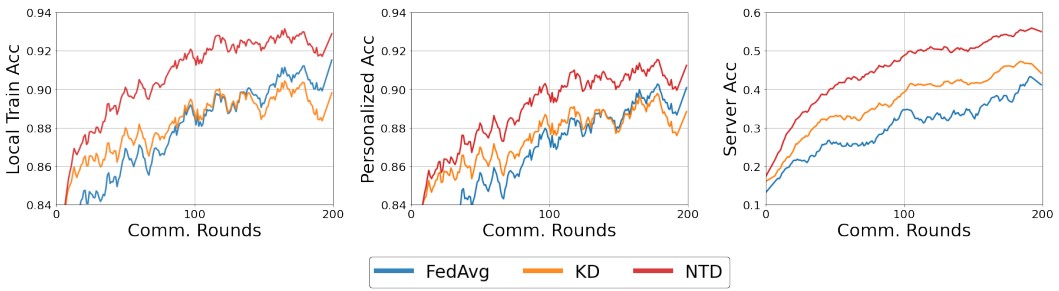

Figure 11: CIFAR-10 (Sharding: $s = 2$) performances from KD and NTD

# I   Personalized performance of FL methods

Here we investigate the personalized performance of our FedNTD. As suggested in Appendix H, although FedNTD aims to improve global convergence, it also improves personalized performance. However, as the learning curves of SCAFFOLD (*the green line*) show, the lower local learning performance does not always lead to the worse server model performance. In all cases, SCAFFOLD shows significantly lower local performance at each round (*the 1st and 2nd row*), but it considerably improves the global convergence (*the 3rd row*).

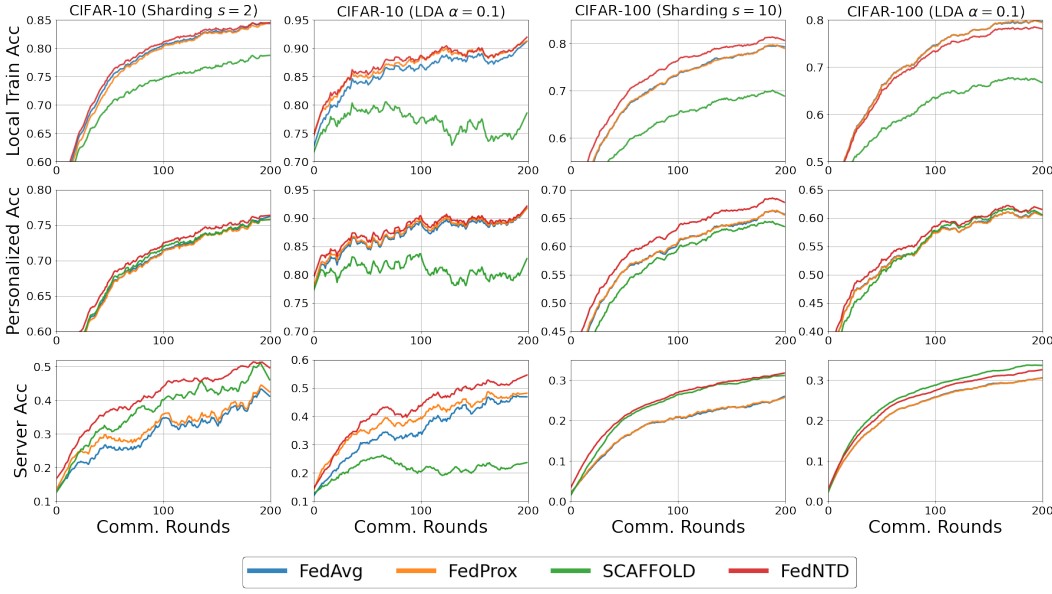

Figure 12: Local and global learning curves of FL methods. The accuracy of the local model is evaluated on: *(Local Train)* - the local private trains samples, and *(Personalized)*: the test samples from the same label distribution with the local client

## J    Comparison to FedAlign

Here we compare our FedNTD with a recently proposed method, FedAlign [38], which shares the motivation of our work that local learning is the bottleneck of FL performance. In FedAlign, a correction term is introduced in the local learning target to obtain local models that generalize well. We implemented our FedNTD on officially released FedAlign code [2], and used the hyperparameters specified in [38]. The results are in Table 13 and Figure 13 shows their corresponding learning curves. In our experiment, although FedAlign improves the performance at some settings (LDA $\alpha = 0.5$), its learning suffers when the heterogeneity level becomes severe. On the other hand, FedNTD consistently improves the performance even in such cases.

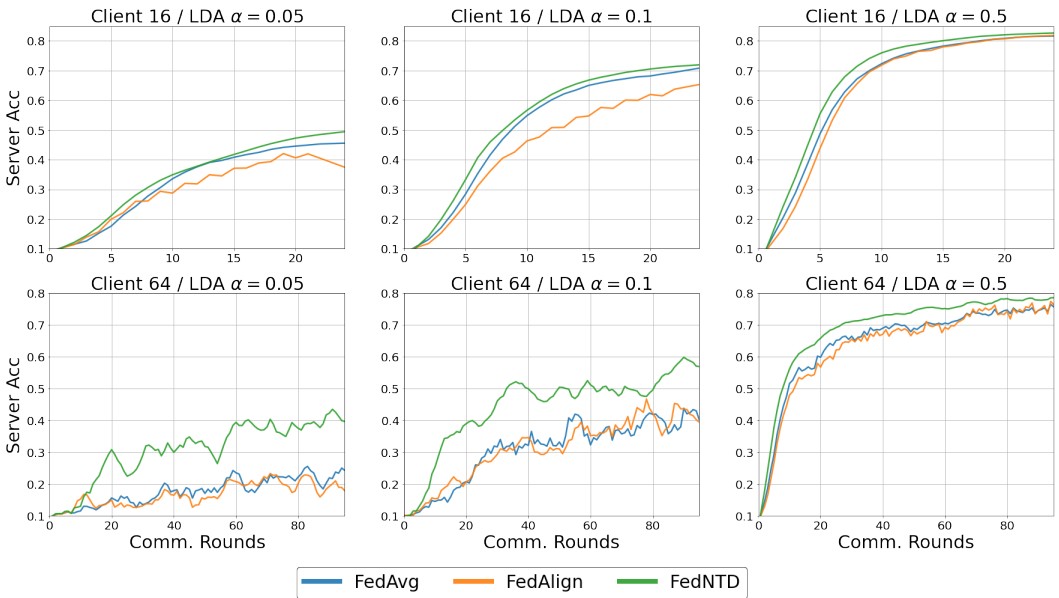

Figure 13: Learning curves that corresponds to Table 13.

Table 13: CIFAR-10 test accuracy. 16 clients participates for each communication round. The number local epochs is 20 for all experiments.

| Client Number (LDA $\alpha$) | FedAvg [37] | FedAlign[38] | FedNTD (ours) |
|---|---|---|---|
| Client 16 ($\alpha = 0.05$) | 0.4556 | 0.3743 | **0.4943** |
| Client 16 ($\alpha = 0.1$) | 0.7083 | 0.6532 | **0.7195** |
| Client 16 ($\alpha = 0.5$) | 0.8163 | 0.8185 | **0.8266** |
| Client 64 ($\alpha = 0.05$) | 0.2535 | 0.1854 | **0.3927** |
| Client 64 ($\alpha = 0.1$) | 0.4247 | 0.3931 | **0.5634** |
| Client 64 ($\alpha = 0.5$) | 0.7568 | 0.7698 | **0.7846** |

The introduced loss term of FedAlign aims to seek out-of-distribution generality w.r.t. global distribution during local training, resulting in the smooth loss landscape across domains (= heterogeneous local distributions in FL context). In Figure 14, we analyze the loss landscape using the parameter perturbation with Gaussian noise and the visualization using top-2 eigenvector axis, as in [38]. Interestingly, our FedNTD also smoothed the local landscape, implying that the local training does not require significant parameter change to fit its local distribution. We expect that one can get insight into the intriguing property in the loss space geometry to tackle the data heterogeneity problem for future work.

---

[2]https://github.com/mmendiet/FedAlign

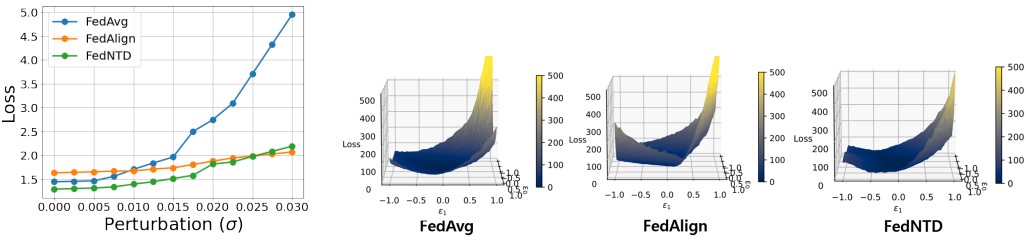

(a) Loss change by perturbation
(b) Visualized loss landscape with Hessian eigenvectors $\epsilon_0$ and $\epsilon_1$.

Figure 14: Loss space of learned model (Client 16 / LDA $\alpha = 0.5$).

## K  Effect of FedNTD Hyperparameters

In Figure 15, we plot the effect of FedNTD hyperparameters on the performance. The result shows that although FedNTD is not much sensitive to the choice of $\beta$, too small $\tau$ significantly drops the accuracy., which may be due to the too stiff not-true probability targets. The effect of both hyperparameters on the forgetting measure $\mathcal{F}$ is in Figure 16.

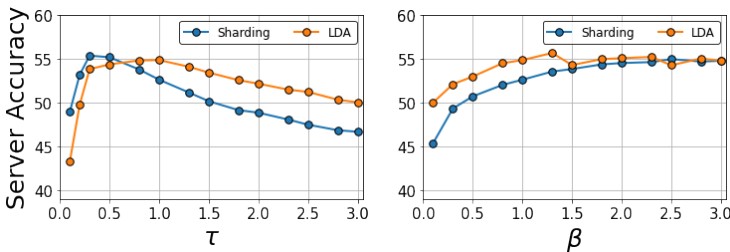

Figure 15: CIFAR-10 (Sharding: $s = 2$, LDA: $\alpha = 0.1$) test accuracy by varying FedNTD hyperparameter $\tau$ and $\beta$ values.

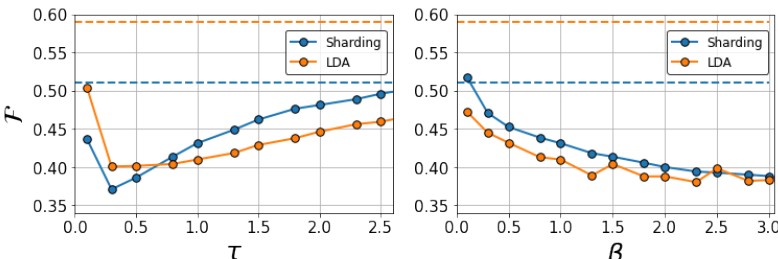

Figure 16: Forgetting $\mathcal{F}$ of FedNTD on CIFAR-10 by varying hyperparameters. The dotted lines stands for the baseline FedAvg.

## L  MSE Loss for Not-True Distillation

We explore how the MSE loss on Not-True Distillation acts. In Table 14, the MSE version FedNTD (FedNTD (MSE)) shows better accuracy and less forgets as $\beta$ grows, but at some degree, the model diverges; thereby cannot reach the original FedNTD, which exploits softmax and KL-Divergence loss to distill the knowledge in the global model. We explain it as matching all not-true logits using MSE logits is too strict to learn the global knowledge since the dark knowledge is mainly contained in top-k logits. FedNTD controls the class signals by using temperature-softened softmax.

Table 14: CIFAR-10 (Sharding s=2) results by varying $\beta$ for FedNTD (MSE) and FedNTD

| Method | FedAvg | FedNTD (MSE) | | | | | | FedNTD |
|---|---|---|---|---|---|---|---|---|
| $\beta$ | 0.0 | 0.001 | 0.005 | 0.01 | 0.05 | 0.1 | 0.3 | 1.0 |
| **Accuracy** | 40.14 | 40.53 | 42.39 | 43.02 | 44.41 | 44.27 | *Failure* | **52.61** |
| **Forgetting** $\mathcal{F}$ | 0.59 | 0.58 | 0.56 | 0.55 | 0.53 | 0.53 | *Failure* | **0.43** |

# M Visualization of Feature Alignment

To analyze feature alignment, we regard a neuron as the basic feature unit and identify individual neuron's class preference as follows:

$$\mathcal{H} = [h_1, h_2 \ldots h_{\mathcal{C}}], \text{ where } h_c = \sum_{i=1}^{N_c} \mathcal{O}(x_{c,i}). \tag{19}$$

Here, the $\mathcal{O}(x_{c,i})$ denotes the neuron's activation on data $x_i$ of class $c$, and $N_c$ is the number of samples for class $c$. For each neuron, we obtain the largest class index $\text{argmax}_i(\mathcal{H}_i)$, to identify the most dominantly encoded class semantics. A similar measure is adopted in [53]. In Figure 17, we visualize the last layer neurons' class preference. In both IID and NIID (Sharding $s = 2$) cases, the features are more well-aligned in FedNTD.

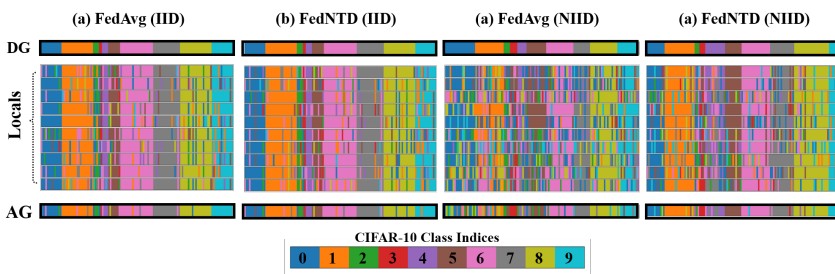

Figure 17: Visualized server test accuracy of FedAvg and FedNTD on CIFAR-10 (Sharding=2).

# N Local features visualization: Hypersphere

To figure out forgetting of knowledge in the global model, we now analyze how the representation on the global distribution changes during local training. To this end, we design a straightforward experiment that shows the change of features on the unit hypersphere. More specifically, we modified the network architecture to map CIFAR-10 (Sharding $s = 2$) input data to 2-dimensional vectors and normalize them to be aligned on the unit hypersphere $S^1 = \{x \in \mathbb{R}^2 : ||x||_2 = 1\}$. We then estimate their probability density function. The global model is learned for 100 rounds of communication on *homogeneous* locals (i.i.d. distributed) and distributed to *heterogeneous* locals with different local distributions. The result is in Figure 18

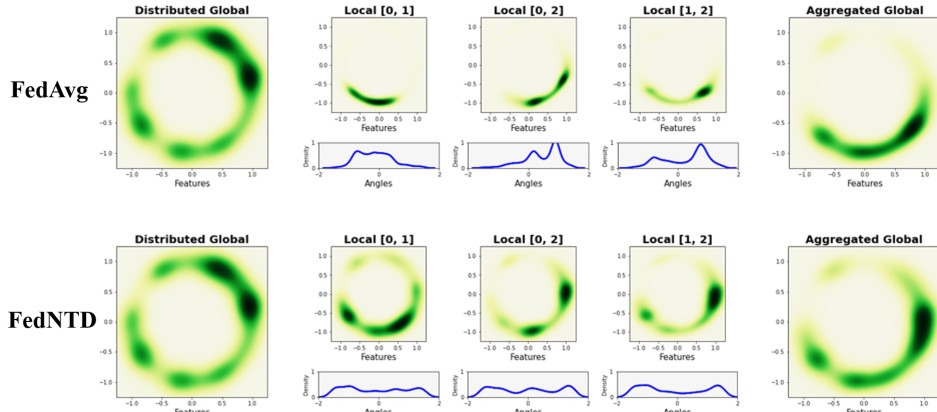

Figure 18: Features of CIFAR-10 (Sharding $s = 2$) test samples on $S^2$. We plot the feature distribution with Gaussian kernel density estimation (KDE) in $\mathbb{R}^2$ and $\arctan(y, x)$ for each point $(x, y) \in S^1$. The distributed global model (first column) is trained on heterogeneous locals (middle 3 columns) and aggregated by parameter averaging (last column).

# O   Local features visualization: T-SNE

We further conduct an additional experiment on features of the trained local model. we trained the global server model for 100 communication rounds on *heterogeneous* (NIID) locals and distributed over 10 *homogeneous* (IID) locals and 10 *heterogeneous* (NIID) locals. In the homogeneous local case (Figure 19a, Figure 20a), the features are clustered by classes, regardless of which local they are learned from. On the other hand, in the heterogeneous local case (Figure 19b, Figure 20b), the features are clustered by which local distribution is learned. In Figure 21, we visualize the effect of FedNTD on the local features.

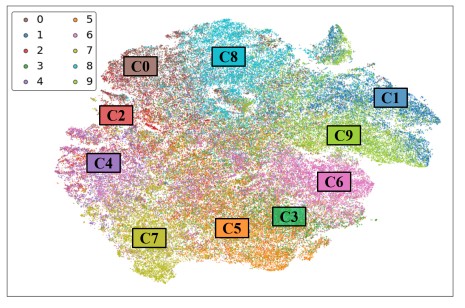

(a) Homogeneous Locals (colored by *classes*)

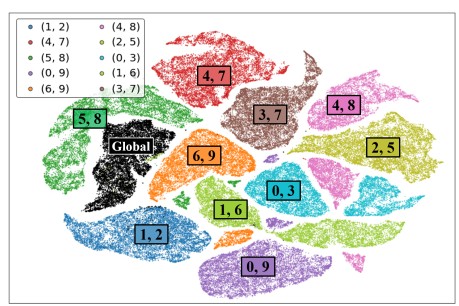

(b) Heterogeneous Locals (colored by *locals*)

Figure 19: T-SNE visualization of features on CIFAR-10 test samples after local training on (a) *homogeneous* local distributions and (b) *heterogeneous* local distributions. The T-SNE is conducted together for the test sample features of global and 10 local models.

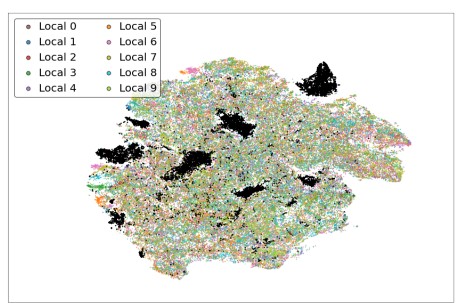

(a) Homogeneous Locals (colored by *classes*)

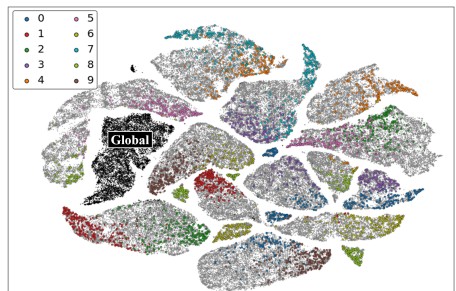

(b) Heterogeneous Locals (colored by *locals*)

Figure 20: T-SNE visualization of feature region shifting on CIFAR-10 test samples after local training on (a) Homogeneous local distributions and (b) heterogeneous local distributions. The T-SNE is conducted together for the test sample features of global and 10 local models.

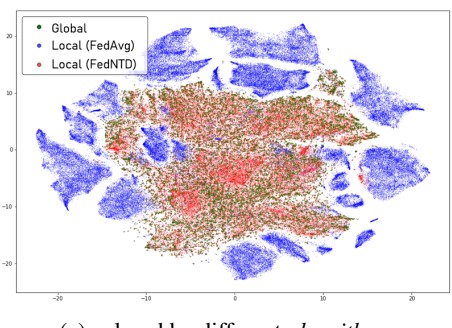

(a) colored by different *algorithms*

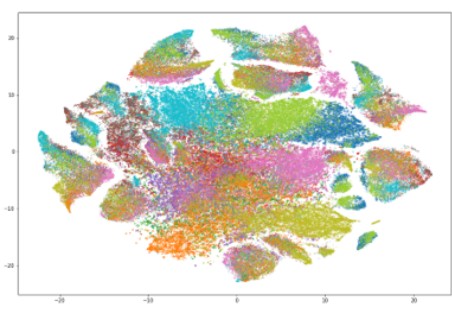

(b) colored by *classes*

Figure 21: T-SNE on CIFAR-10 testset samples fter local training on *heterogeneous* local distributions by **FedAvg** and **FedNTD**. The T-SNE is conducted together for the test sample features of global and 20 local models (10 for FedAvg and 10 for FedNTD).

# P   Proof of Proposition 1

*Proof.* Since the class-wise gradient $g_i$ are mutually orthogonal and have uniform weight, from the unitarily invariance of the 2-norm we have:

$$\Lambda(\beta) = \frac{\frac{1}{K}\sum_{k=1}^{K}\|p^k + \beta\tilde{p}^k\|^2}{\|\sum_{k=1}^{K}p^k + \beta\tilde{p}^k\|^2} \overset{(\spadesuit)}{=} \frac{1}{K}\frac{\sum_{k=1}^{K}\sum_{c=1}^{\mathcal{C}}(p^k(c) + \beta\tilde{p}^k(c))^2}{\sum_{c=1}^{\mathcal{C}}(\sum_{k=1}^{K}p^k(c) + \beta\tilde{p}^k(c))^2} \tag{20}$$

$$\overset{(\clubsuit)}{=} \frac{1}{(1+\beta)^2}\frac{\mathcal{C}}{K^3}\sum_{k=1}^{K}\sum_{c=1}^{\mathcal{C}}(p^k(c) + \beta\tilde{p}^k(c))^2 \tag{21}$$

$$= \frac{1}{(1+\beta)^2}\frac{\mathcal{C}}{K^3}\sum_{k=1}^{K}\mathcal{C}\,\mathbb{E}_{c\in[\mathcal{C}]}[(p^k + \beta\tilde{p}^k)^2] = \frac{1}{(1+\beta)^2}\frac{\mathcal{C}^2}{K^3}\sum_{k=1}^{K}\mathbb{E}_{c\in[\mathcal{C}]}[(p^k + \beta\tilde{p}^k)^2] \tag{22}$$

$$= \frac{1}{(1+\beta)^2}\frac{\mathcal{C}^2}{K^3}\sum_{k=1}^{K}\left(\text{Var}_{c\in[\mathcal{C}]}[p^k + \beta\tilde{p}^k] + (1+\beta)^2\right) \tag{23}$$

$$= \frac{1}{(1+\beta)^2}\frac{\mathcal{C}^2}{K^3}\sum_{k=1}^{K}\left(\text{Var}_{c\in[\mathcal{C}]}\left[p^k + \beta\frac{1-p^k}{\mathcal{C}-1}\right]\right) + \frac{\mathcal{C}^2}{K^2} \tag{24}$$

$$= \frac{1}{(1+\beta)^2}\frac{\mathcal{C}^2}{K^3}\sum_{k=1}^{K}\left((1-\frac{\beta}{\mathcal{C}-1})^2\text{Var}_{c\in[\mathcal{C}]}[p^k]\right) + \frac{\mathcal{C}^2}{K^2}\,. \tag{25}$$

Where the $(\spadesuit)$ follows from $p^k = \sum_{c=1}^{\mathcal{C}}p^k(c)\mathbf{e}_k(\in \Delta_{\mathcal{C}})$, and $(\clubsuit)$ holds because we are assuming uniform global data distribution. That is, we have the following equation from the symmetry over classes.

$$\sum_{k=1}^{K}p^k(c) = \sum_{k=1}^{K}\tilde{p}^k(c) = \frac{K}{\mathcal{C}}\,. \tag{26}$$

By differentiating the equation (25), we have:

$$\frac{\partial\Lambda(\beta)}{\partial\beta} = \left(\frac{\mathcal{C}^2}{K^3(\mathcal{C}-1)^2}\sum_{k=1}^{K}\text{Var}_{c\in[\mathcal{C}]}[p^k]\right)\frac{\partial}{\partial\beta}\frac{(\mathcal{C}-(1+\beta))^2}{(1+\beta)^2} \tag{27}$$

$$= \left(\frac{\mathcal{C}^2}{K^3(\mathcal{C}-1)^2}\sum_{k=1}^{K}\text{Var}_{c\in[\mathcal{C}]}[p^k]\right)\left(-2\frac{\mathcal{C}^2}{(1+\beta)^3} + 2\frac{\mathcal{C}}{(1+\beta)^2}\right) \tag{28}$$

$$= -\left(\frac{2\mathcal{C}^3}{K^3(\mathcal{C}-1)^2}\sum_{k=1}^{K}\text{Var}_{c\in[\mathcal{C}]}[p^k]\right)\left(\frac{\mathcal{C}}{(1+\beta)^3} - \frac{1}{(1+\beta)^2}\right)\,. \tag{29}$$

$\square$

By defining $M_{K,\mathcal{C},p} > 0$ in the first bracket, we have:

$$\frac{\partial\Lambda}{\partial\beta} = -M_{K,\mathcal{C},p}\left(\frac{\mathcal{C}}{(1+\beta)^3} - \frac{1}{(1+\beta)^2}\right)\,, \tag{30}$$

for all $\beta \geqslant 0$. If $\beta \leqslant \mathcal{C}/2 - 1$, we have $\mathcal{C}/(1+\beta) \geqslant 2$, and get following desired inequality:

$$\frac{\partial\Lambda}{\partial\beta} \leqslant -M_{K,\mathcal{C},p}\frac{1}{(1+\beta)^2}\,. \tag{31}$$

# Q   Proof of Proposition 2

*Proof.* First, we show the first equation. The summation for the true class is:

$$\mathcal{L}_{\text{KL}}^{\text{true}} = -\frac{1}{N}\sum_{i=1}^{N}q_\tau^{g,i}(y_i)\log\left[\frac{q_\tau^{l,i}(y_i)}{q_\tau^{g,i}(y_i)}\right] \tag{32}$$

Note that $\sum_{i=1}^{N} = \sum_{c=1}^{\mathcal{C}} \sum_{i \in \mathcal{S}_c}$ and $i \in \mathcal{S}_c \Rightarrow y_i = c$. By using these, we get:

$$\mathcal{L}_{\text{KL}}^{\text{true}} = -\frac{1}{N} \sum_{i=1}^{N} q_\tau^{g,i}(y_i) \log \left[ \frac{q_\tau^{l,i}(y_i)}{q_\tau^{g,i}(y_i)} \right] = -\frac{1}{N} \sum_{c=1}^{\mathcal{C}} \sum_{i \in \mathcal{S}_c} q_\tau^{g,i}(c) \log \left[ \frac{q_\tau^{l,i}(c)}{q_\tau^{g,i}(c)} \right] \tag{33}$$

$$= -\sum_{c=1}^{\mathcal{C}} \boldsymbol{p_c} \cdot \left( \sum_{i \in \mathcal{S}_c} \frac{1}{|\mathcal{S}_c|} q_\tau^{g,i}(c) \log \left[ \frac{q_\tau^{l,i}(c)}{q_\tau^{g,i}(c)} \right] \right) \tag{34}$$

$$= -\sum_{c=1}^{\mathcal{C}} \boldsymbol{p_c} \cdot \mathbb{E}_{i \in \mathcal{S}_c} \left[ q_\tau^{g,i}(c) \log \left[ \frac{q_\tau^{l,i}(c)}{q_\tau^{g,i}(c)} \right] \right]. \tag{35}$$

Next, we derive the not-true part of the Kullback-Leibler divergence:

$$-N\mathcal{L}_{\text{KL}}^{\text{not-true}} = \sum_{i=1}^{N} \sum_{c' \neq y_i}^{\mathcal{C}} q_\tau^{g,i}(c') \log \left[ \frac{q_\tau^{l,i}(c')}{q_\tau^{g,i}(c')} \right]. \tag{36}$$

By using the double summation technique ($\star$), we have:

$$-N\mathcal{L}_{\text{KL}}^{\text{not-true}} = \sum_{c=1}^{\mathcal{C}} \sum_{i \in \mathcal{S}_c} \sum_{c' \neq c}^{\mathcal{C}} q_\tau^{g,i}(c') \log \left[ \frac{q_\tau^{l,i}(c')}{q_\tau^{g,i}(c')} \right] = \sum_{c=1}^{\mathcal{C}} \sum_{c' \neq c}^{\mathcal{C}} \sum_{i \in \mathcal{S}_c} q_\tau^{g,i}(c') \log \left[ \frac{q_\tau^{l,i}(c')}{q_\tau^{g,i}(c')} \right] \tag{37}$$

$$\overset{(\star)}{=} \sum_{c'=1}^{\mathcal{C}} \sum_{c \neq c'}^{\mathcal{C}} \sum_{i \in \mathcal{S}_c} q_\tau^{g,i}(c') \log \left[ \frac{q_\tau^{l,i}(c')}{q_\tau^{g,i}(c')} \right] = \sum_{c=1}^{\mathcal{C}} \sum_{c' \neq c}^{\mathcal{C}} \sum_{i \in \mathcal{S}_{c'}} q_\tau^{g,i}(c) \log \left[ \frac{q_\tau^{l,i}(c)}{q_\tau^{g,i}(c)} \right] \tag{38}$$

$$= (\mathcal{C} - 1) \sum_{c=1}^{\mathcal{C}} \frac{\sum_{c' \neq c} |\mathcal{S}_{c'}|}{\mathcal{C} - 1} \left( \sum_{c' \neq c} \frac{1}{\sum_{c' \neq c} |\mathcal{S}_{c'}|} \sum_{i \in \mathcal{S}_{c'}} q_\tau^{g,i}(c) \log \left[ \frac{q_\tau^{l,i}(c)}{q_\tau^{g,i}(c)} \right] \right) \tag{39}$$

$$= (\mathcal{C} - 1) \sum_{c=1}^{\mathcal{C}} N \tilde{\boldsymbol{p}}_{\boldsymbol{c}} \cdot \mathbb{E}_{i \notin \mathcal{S}_c} \left[ q_\tau^{g,i}(c) \log \left[ \frac{q_\tau^{l,i}(c)}{q_\tau^{g,i}(c)} \right] \right]. \tag{40}$$

$\square$

Therefore, we get our desired result:

$$\frac{\mathcal{L}_{\text{KL}}^{\text{not-true}}}{\mathcal{C} - 1} = \sum_{c=1}^{\mathcal{C}} \tilde{\boldsymbol{p}}_{\boldsymbol{c}} \mathbb{E}_{i \notin \mathcal{S}_c} \left[ q_\tau^{g,i}(c) \log \left[ \frac{q_\tau^{l,i}(c)}{q_\tau^{g,i}(c)} \right] \right]. \tag{41}$$

## R Derivation of Equation 15

*Proof.* The main part of the proof is well-known inequality for smooth functions, which is derived from the Taylor approximation. Since $\mathcal{L}_i : \mathcal{W} \subset \mathbb{R}^n \to \mathbb{R}$ is smooth function, we have

$$\mathcal{L}_i(w) = \mathcal{L}_i(w_i) + \nabla \mathcal{L}_i(w_i) \cdot (w - w_i) + \int_0^1 (1-t)(w-w_i)^\top \cdot \nabla^2 \mathcal{L}_i(w_i + t(w - w_i)) \cdot (w - w_i) dt \tag{42}$$

$$= \mathcal{L}_i(w_i) + \int_0^1 (1-t)(w-w_i)^\top \cdot \nabla^2 \mathcal{L}_i(w_i + t(w - w_i)) \cdot (w - w_i) dt \tag{43}$$

$$\leqslant \mathcal{L}_i(w_i) + \lambda \int_0^1 (1-t)(w-w_i)^\top \cdot (w - w_i) dt \qquad (\nabla^2 \mathcal{L}_i(w) \preceq \lambda)$$

$$= \mathcal{L}_i(w_i) + \frac{\lambda}{2} \|w - w_i\|^2. \tag{44}$$

$\square$

# S Proof of Proposition 3

*Proof.* To show this corollary, enough to show that the below minimax problem is attained on the uniform distribution.

$$\inf_{p \in \Delta_C} \sup_{\mathbb{P} \in \Pi} \mathbb{E}_{p' \sim \mathbb{P}}[\|p' - p\|]. \tag{45}$$

Let us define $p \mapsto \sup_{\mathbb{P} \in \Pi} \mathbb{E}_{p' \sim \mathbb{P}}[\|p' - p\|]$ as $F(p)$. First, we check the continuity of $F$. That is:

$$|F(p_2) - F(p_1)| \leqslant \left| \sup_{\mathbb{P} \in \Pi} \mathbb{E}_{p' \sim \mathbb{P}}[\|p' - p_2\|] - \sup_{\mathbb{P} \in \Pi} \mathbb{E}_{p' \sim \mathbb{P}}[\|p' - p_1\|] \right| \tag{46}$$

$$\leqslant \sup_{\mathbb{P} \in \Pi} \left| \mathbb{E}_{p' \sim \mathbb{P}}[\|p' - p_2\| - \|p' - p_1\|] \right| \leqslant \sup_{\mathbb{P} \in \Pi} \mathbb{E}_{p' \sim \mathbb{P}}[|\|p' - p_2\| - \|p' - p_1\||] \tag{47}$$

$$\leqslant \sup_{\mathbb{P} \in \Pi} \mathbb{E}_{p' \sim \mathbb{P}}[\|p_1 - p_2\|] \leqslant \|p_1 - p_2\|. \tag{48}$$

Therefore, since the function $F$ is 1-Lipschitz, it is clearly continuous. Now, since $\Delta_C$ is compact, we have a minimizer $p_0 \in \Delta_C$ of above minimax value. Since norm and expectation is convex function, $F$ is convex. Therefore, for arbitrary minimizer $p_0$ and cycle $\sigma = (1\, 2\, \cdots\, C) \in \mathcal{S}_C$, we have:

$$F(\text{unif. dist}) = F\left( \frac{1}{C} \sum_i \sigma^i(p_0) \right) \leqslant \frac{1}{C} \sum_i F(\sigma^i(p_0)). \tag{49}$$

Now, we argue that $F(\sigma^i(p_0)) = F(p_0)$. From the definition of $F$,

$$F(\sigma^i(p_0)) = \sup_{\mathbb{P} \in \Pi} \mathbb{E}_{p' \sim \mathbb{P}}[\|p' - \sigma^i(p_0)\|] = \sup_{\mathbb{P} \in \Pi} \mathbb{E}_{p' \sim \mathbb{P}}[\|\sigma^i(\sigma^{-i}(p')) - \sigma^i(p_0)\|] \tag{50}$$

$$= \sup_{\mathbb{P} \in \Pi} \mathbb{E}_{p' \sim \mathbb{P}}[\|\sigma^i(\sigma^{-i}(p')) - \sigma^i(p_0)\|] \tag{51}$$

$$= \sup_{\mathbb{P} \in \Pi} \int_{\Delta_C} \|\sigma^i(\sigma^{-i}(p')) - p_0\| \, d\mathbb{P}(p') \tag{52}$$

$$= \sup_{\mathbb{P} \in \Pi} \int_{\Delta_C} \|\sigma^{-i}(p') - p_0\| \, d\mathbb{P}(\sigma^i(\sigma^{-i}(p'))) \tag{53}$$

$$= \sup_{\mathbb{P} \in \Pi} \int_{\Delta_C} \|p'' - p_0\| \, d\mathbb{P}(\sigma^i(p'')) \tag{54}$$

$$= \sup_{\mathbb{P} \in \Pi} \int_{\Delta_C} \|p'' - p_0\| \, d\mathbb{P}(p'') = F(p_0). \qquad (\Pi \text{ is } \mathcal{S}_C\text{-invariant } (\sigma^i \in \mathcal{S}_C)) \tag{55}$$

From the equation equation (49), we have:

$$F(\text{unif. dist}) \leqslant \frac{1}{C} \sum_i F(\sigma^i(p_0)) = \frac{1}{C} \sum_i F(p_0) = F(p_0). \tag{56}$$

Since $p_0$ is minimizer, we can argue that the uniform distribution also attains the minimum. $\square$