# OpenReview forum: "Preservation of the Global Knowledge by Not-True Distillation in Federated Learning"
_NeurIPS.cc/2022/Conference — NeurIPS 2022 Accept_

### Official Review · Reviewer_qrmZ · 2022-07-09

**Rating:** 5
**Confidence:** 3
**Soundness:** 3 good
**Presentation:** 3 good
**Contribution:** 3 good

**Summary:**

The paper proposes a method to mitigate catastrophic forgetting in federated learning, which focuses on improving the performance of the local model by teaching the knowledge of the global model to the local model through an improved knowledge distillation method. According to the experimental results presented in the paper, the method possesses a good performance.

**Questions:**

1. For the MOON method, the results in the table seem to be not as good as the performance of the original paper. the MOON method is more sensitive to parameters, and it is suggested that more comparisons of different hyper parameters can be put in the appendix

2. In fact, I am curious why discarding not-true has a positive impact, can you describe it simply?

**Limitations:**

See the weaknesses and questions

**Strengths And Weaknesses:**

**Strengths**
1. The experimental setup of this paper is relatively adequate, taking into account scenarios with different federal settings
2. The experimental approach is simple and achieves good results
3. The method is well analyzed

**Weaknesses**
1. lack of comparison. The following paper is similar in approach to the method proposed in this paper, and also performs knowledge distillation locally
    - Mendieta M, Yang T, Wang P, et al. Local Learning Matters: Rethinking Data Heterogeneity in Federated Learning[C]//Proceedings of the IEEE/CVF Conference on Computer Vision and Pattern Recognition. 2022: 8397-8406.

---

> ### Author Response · Authors · 2022-08-02
> **Q2) Effect of discarding true-class in FedNTD**
>
> **(Short Summary)**
>
> Discarding the true-class logits during distillation makes the true-class signal only comes from the labeled data in the local client by the Cross-Entropy Loss. In this way, the distillation does not hinder attaining the new knowledge on the local data while obligating the distillation loss focus on preserving the global model’s knowledge on the locally not available classes.
>
> **(Long Details)**
>
> In our paper, “Forgetting” is defined as the accuracy drop in the classes that the global model predicts well. As the local model fits only on the locally available data, the trained local models perform very well on the in-local distribution ($p(D)$ in our paper) but significantly drop their performance on the out-local distribution ($\tilde{p}(D)$ in our paper). Our main message is that FL performance is decided by balancing the learning of the new knowledge on $p(D)$ and preserving the old knowledge on $\tilde{p}(D)$.
>
> In Proposition 2, we suggest that distillation on true-class logits corresponds to $p(D)$ and not-true class logits correspond to $\tilde{p}(D)$. As the role of distillation loss is preserving old knowledge on $\tilde{p}(D)$, we discard the true-class logits, and now the learning on $p(D)$ is only performed by the CE Loss using the ground-truth labels. In a nutshell, such discarding provides a better trade-off between attaining new knowledge and preserving old knowledge.
>
> Then, a natural question is whether the performance is actually hindered by including true-class logits in distillation loss. Our answer is YES, and we provide the comparison experiment in **Appendix I**: Comparison to KD. We show that NTD significantly outperforms the vanilla KD, and the performance is consistently improved as the distillation loss function reaches from vanilla KD to NTD. The below table is from **Appendix I**, which shows how the performance improves as the distillation loss $L(\lambda) = (1-\lambda) \cdot {L_{KD}} + \lambda \cdot {L_{NTD}}$ reaches from $L_{KD}$ to $L_{NTD}$ by increasing $\lambda$.
>
> | **Partition Method** | **0.0 (KD)** | **0.1** | **0.3** | **0.95** | **0.7** | **0.9** | **1.0 (NTD)** |
> |:--------------------:|:------------:|:-------:|:-------:|:-------:|:-------:|:-------:|:-------------:|
> | Sharding (s=2) |     46.2     |   46.4  |   46.9  |   47.6  |   48.8  |   50.2  |    **52.6**   |
> | LDA ($\alpha$ = 0.1) |     50.8     |   50.9  |   51.4  |   51.9  |   52.6  |   53.6  |    **54.9**   |
>
> **Table: CIFAR-10 test accuracy by varying distillation loss function.**
>
> Moreover, the train accuracy and personalized accuracy of the local model is significantly improved, which supports that the NTD tackles the hindered local learning when using KD. Please see more details in the supplementary material which includes the full learning curves.
>
> |    **KD**   | **Round 50** | **Round 100** | **Round 200** |   **NTD**   | **Round 50** | **Round 100** | **Round 200** |
> |:-----------:|:------------:|:-------------:|:-------------:|:-----------:|:------------:|:-------------:|:-------------:|
> | Local Train |    0.8580    |     0.8812    |     0.8854    | Local Train |    0.8876    |     0.915     |     0.9186    |
> |  Local Test |    0.8472    |     0.8704    |     0.8767    |  Local Test |    0.8648    |     0.8900    |     0.8954    |
> | Server Test |    0.3310    |     0.4060    |     0.4642    | Server Test |    0.3859    |     0.4945    |     0.5723    |
>
> **Table: CIFAR-10 Local and server performance at the intermediate rounds.**
>
> We believe that a concurrent work Decoupled Knowledge Distillation (DKD)[1], which studied logits decoupling in a general centralized training scenario, may provide further insight into the role of NTD in FL. It suggests that the amount of not-true class signals is restricted by the model’s prediction confidence on true-class in standard KD. They suggested to decouple knowledge distillation for the true-class and not-true classes by using separated hyperparameters to overcome this issue.
>
> However, such restriction by distillation loss on true-class is even more significant in the FL scenario. With conventional FL algorithms, we can observe a considerable drop of out-local class accuracies after each local update, which is forgetting. Discarding true-class is a remedy for the forgetting issue since it can retain the out-local information regardless of the global model’s prediction of the true-class. However, discarding is more adequate than decoupling in FL since the global model’s prediction may be noisy during the learning process, which is different from the centralized distillation scenario that generally assumes a well pre-trained teacher model.
>
> [1]. [Zhao, Borui, et al. “Decoupled Knowledge Distillation.” Proceedings of the IEEE/CVF Conference on Computer Vision and Pattern Recognition. 2022.](https://openaccess.thecvf.com/content/CVPR2022/html/Zhao_Decoupled_Knowledge_Distillation_CVPR_2022_paper.html).

---

> ### Author Response · Authors · 2022-08-02
> **Q1) Performance of MOON with different hyperparameters**
>
> For the implemented algorithms, the hyperparameters which perform **best** are chosen from the searched candidates. In our supplementary material, We provide the searched hyperparameter candidates in **Appendix C.4**: Algorithm Implementation Details.
>
> We emphasize that we used the fixed hyperparameters for our FedNTD as the default values (β= 1, τ = 1) for all experiments to show a more intuitive comparison. However, hyperparameter choice can further improve the accuracy, as suggested in **Appendix L**: Effect of FedNTD Hyperparameters.
>
> MOON[1] is one of the SOTA algorithms to mitigate data heterogeneity in FL. However, even the state-of-the-art methods perform well only in specific setups, and their performance often deteriorates below FedAvg, as reported in [2]. For example, another SOTA method SCAFFOLD[3] significantly underperforms FedAvg in the experiments reported in [1].
>
> [1] [Li, Qinbin, Bingsheng He, and Dawn Song. "Model-contrastive federated learning." Proceedings of the IEEE/CVF Conference on Computer Vision and Pattern Recognition. 2021.](https://openaccess.thecvf.com/content/CVPR2021/html/Li_Model-Contrastive_Federated_Learning_CVPR_2021_paper.html)
>
> [2] [Li, Qinbin, et al. "Federated learning on non-iid data silos: An experimental study." arXiv preprint arXiv:2102.02079 (2021).](https://arxiv.org/abs/2102.02079)
>
> [3] [Karimireddy, Sai Praneeth, et al. "Scaffold: Stochastic controlled averaging for federated learning." International Conference on Machine Learning. PMLR, 2020.](http://proceedings.mlr.press/v119/karimireddy20a.html)

---

> ### Author Response · Authors · 2022-08-02
> **Weakness 1) Lack of Comparison**
>
> We appreciate the reviewer for introducing a paper that shares similar motivation with our work. It seems that the “local learning generality” in the suggested paper [1] is closely related to “global knowledge preservation” in our work. Here we first provide our comparison experiment and discuss some intriguing properties of FedNTD in the sense of FedAlign.
>
> The table below is the comparison between FedNTD and FedAlign. For a clear comparison, we implemented our FedNTD on the officially released code of FedAlign. All the hyperparameters are set as specified in [1]. We could find that FedAlign outperforms in some settings used in the paper (ex. $\alpha=0.5$), but FedNTD performs much better in severe heterogeneity levels (at smaller $\alpha$) and shows more stable convergence. Our supplementary material includes the detailed explanation in **Appendix K**: Comparison to FedAlign.
>
> | Client Number / LDA $\alpha$ | **FedAvg** | **FedAlign** | **FedNTD** |
> |:----------------------------:|:----------:|:------------:|:----------:|
> |      Client 16 / $\alpha=0.05$      |   0.4556   |    0.3743    |  **0.4943**  |
> |       Client 16 / $\alpha=0.1$      |   0.7083   |    0.6532    |   **0.7195**  |
> |       Client 16 / $\alpha=0.5$     |   0.8163   |    0.8185    |   **0.8266**   |
> |      Client 64 /$\alpha=0.05$      |   0.2535   |    0.1854    |   **0.3927**  |
> |       Client 64 / $\alpha=0.1$      |   0.4247   |    0.3931    |   **0.5634**  |
> |       Client 64 / $\alpha=0.5$      |   0.7568   |    0.7698    |   **0.7846**  |
>
> **Table 1: Comparison to FedAlign on CIFAR-10**
>
> To best our understanding, FedAlign aims to seek out-of-distribution generality w.r.t. global distribution during local training, resulting in the smooth loss landscape across domains (= heterogeneous local distributions in FL context). Interestingly, our FedNTD also smoothed the local landscape, implying that the local training does not require significant parameter change to fit its local distribution. The table below shows the loss change when the weights are perturbed by gaussian noise with the standard deviation $\sigma$. In our revised supplementary material, we include the plot and visualized loss landscape in **Appendix K**: Comparison to FedAlign. We expect that one can get insight into the intriguing property in the loss space geometry to tackle the data heterogeneity problem for future work.
>
> | **Method / Perturbation (σ)** |  **0.0**  | **0.005** |  **0.01** | **0.015** |  **0.02** | **0.025** |  **0.03** |
> |:---------------------------------:|:---------:|:---------:|:---------:|:---------:|:---------:|:---------:|:---------:|
> |             **FedAvg**            |   1.404   |   1.609   |   1.387   |   2.140   |   2.154   |   3.101   |   5.361   |
> |            **FedAlign**           |   1.639   |   1.652   |   1.672   |   1.649   |   1.806   | **2.066** | **2.042** |
> |             **FedNTD**            | **1.283** | **1.322** | **1.307** | **1.588** | **1.694** |   2.585   |   2.149   |
>
> **Table 2: Perturbed server model loss (Client 16 / $\alpha=0.5$).**
>
> [1]. [Mendieta, Matias, et al. "Local Learning Matters: Rethinking Data Heterogeneity in Federated Learning." Proceedings of the IEEE/CVF Conference on Computer Vision and Pattern Recognition. 2022.](https://openaccess.thecvf.com/content/CVPR2022/html/Mendieta_Local_Learning_Matters_Rethinking_Data_Heterogeneity_in_Federated_Learning_CVPR_2022_paper.html)

---

### Official Review · Reviewer_sAmP · 2022-07-10

**Rating:** 4
**Confidence:** 4
**Soundness:** 3 good
**Presentation:** 3 good
**Contribution:** 2 fair

**Summary:**

This paper suggests that forgetting could be the bottleneck of global convergence and then proposes FedLSD to handle the forgetting problem by using Not-True Distillation in the local training stage. Unlike traditional distillation methods in Federated Learning, FedLSD does not require additional auxiliary datasets because the distillation works on the clients. To preserve global knowledge on out-local distribution, the distillation only considers the not-true classes. Some experiments are conducted to show its effectiveness.

**Questions:**

See Weaknesses 2 and 4.

**Limitations:**

1. The novelty of the paper is limited.
2. Distillation on the client side may limit the model's learning from local data on the clients

**Strengths And Weaknesses:**

**Strengths**
1. Knowledge forgetting in the global model is a key issue. This paper makes a lot of analysis on forgetting in federated learning.
2. The main strength of the algorithm is that there is no need to use additional auxiliary data during distillation.
3. This paper is well written and easy to read.

**Weaknesses**
1. The knowledge distillation used on the clients is straightforward, although the author expands it to not-true distillation, the novelty is still limited.
2. In the experiment section, direct distillation on clients should also be used as a baseline; otherwise, one cannot see how much impact the proposed Not-True Distillation brings to the results.
3. Figure 1 is less informative. Any method based on regularization can be expressed as Figure 1(c), so it is hard to see the characteristics of the proposed method.
4. According to previous observations and experience, distillation on the client side is very likely to limit the model's learning from local data on the clients. Although this paper only uses the Not-True classes for distillation, such a problem may still exist to restrict the learning of new knowledge.

---

> ### Author Response · Authors · 2022-08-02
> **Weakness 4) Limits on learning local data by client-side distillation**
>
> As the reviewer pointed out, the main reason that we discard true-class logits in the NTD is not to restrict learning from local data when using the distillation loss. The reviewer is concerned whether the local learning is still limited when using NTD loss in local training.
>
> However, we observe that the performance on local distribution with NTD loss is even better than without it (**Figure 6** and **Figure 7**. in our paper). In **Section 5** of our paper, we suggest that FedNTD reduces the required learning distance, which we defined as the distance between global model parameters and learned local model parameters. (**Proposition 3** and its empirical validation in **Figure 8**.)
>
> In other words, it could be possible that the local learning for a single client could be restricted by the regularized training. However, when we consider multiple clients/communication rounds, FedNTD rather benefits local learning under the limited number of local epochs by distributing a better initial learning point, the global model.
>
> As shown in the below table, FedNTD effectively improves local learning when compared to the KD.
>
> |    **KD**   | **Round 50** | **Round 100** | **Round 200** |   **NTD**   | **Round 50** | **Round 100** | **Round 200** |
> |:-----------:|:------------:|:-------------:|:-------------:|:-----------:|:------------:|:-------------:|:-------------:|
> | **Local Train** |    0.8580    |     0.8812    |     0.8854    | **Local Train** |    0.8876    |     0.915     |     0.9186    |
> |  **Local Test (Personalized)** |    0.8472    |     0.8704    |     0.8767    |  **Local Test (Personalized)** |    0.8648    |     0.8900    |     0.8954    |
>
> **Table 1: CIFAR-10 (Sharding s=2) Local Model Performances**
>
> Although personalized performance is outside the scope of our work, we expect that it could help address the reviewer's concerns about restricted local learning. The table below shows the local performances at the intermediate communication rounds. The FedNTD shows considerable improvement in the local model's personalized performance.
>
> | **Personalized Acc** | **Round 50** | **Round 100** | **Round 200** | **Server Acc** | **Round 50** | **Round 100** | **Round 200** |
> |:--------------------:|:------------:|:-------------:|:-------------:|:--------------:|:------------:|:-------------:|:-------------:|
> |      **FedAvg**      |    0.6587    |     0.7218    |     0.7499    |   **FedAvg**   |    0.2382    |     0.3810    |     0.4014    |
> |      **FedProx**     |    0.6590    |     0.7208    |     0.7448    |   **FedProx**  |    0.2684    |     0.4012    |     0.4148    |
> |     **SCAFFOLD**     |    0.6680    |   **0.7306**  |     0.7479    |  **SCAFFOLD**  |    0.2894    |     0.4421    |     0.4460    |
> |      **FedNTD**      |  **0.6736**  |     0.7294    |   **0.7537**  |   **FedNTD**   |  **0.3548**  |   **0.4502**  |   **0.5261**  |
>
> **Table 2: CIFAR-10 Personalized performance of FL algorithms.**
>
> Please see more details, including full learning curves on other settings, in **Appendix J**: Personalized performance of FL methods in our revised supplementary material.

---

> ### Author Response · Authors · 2022-08-02
> **Weakness 3) Informativeness of Figure 1.**
>
> As the reviewer pointed out, any regularization method that preserves global knowledge during local training can be expressed in **Figure 1 (c).** We illustrated **Figure 1 (a)** and **Figure 1 (b)** to show the analogy between continual learning and federated learning. In **Figure (c)**, we tried to deliver that the federated learning performance also can be improved if we develop an adequate method to prevent forgetting.
>
> However, we admit that the readers could be confused if they understand it as if it is a conceptualized illustration of the proposed method. We will refine the figure in the camera-ready version.

---

> ### Author Response · Authors · 2022-08-02
> **Weakness 2) Comparison to direct distillation baseline**
>
> Although we could not include it in the main paper due to the page limit, the comparison to the direct distillation (KD) is included in **Appendix I**: Comparison to KD. To summarize, NTD significantly improves the performance over the direct distillation.
>
> The table below shows how the performance improves as the distillation loss $L(\lambda) = (1-\lambda) \cdot {L_{KD}} + \lambda \cdot {L_{NTD}}$ reaches from $L_{KD}$ to $L_{NTD}$ by increasing $\lambda$.
>
> | **Partition Method** | **0.0 (KD)** | **0.1** | **0.3** | **0.5** | **0.7** | **0.9** | **1.0 (NTD)** |
> |:--------------------:|:------------:|:-------:|:-------:|:-------:|:-------:|:-------:|:-------------:|
> | Sharding (s=2) |     46.2     |   46.4  |   46.9  |   47.6  |   48.8  |   50.2  |    **52.6**   |
> | LDA ($\alpha$ = 0.1) |     50.8     |   50.9  |   51.4  |   51.9  |   52.6  |   53.6  |    **54.9**   |
>
> **Table 1: CIFAR-10 test accuracy by varying $\lambda$.**

---

> ### Author Response · Authors · 2022-08-02
> **Weakness 1) Novelty of the paper**
>
> To best our understanding, the reviewer is concerned about whether discarding true-class logits in distillation is novel enough in FL literature. Please consider the following factors when considering our work's novelty.
>
> At first, our forgetting view using the class-wise approach is novel in FL literature, which we formally analyze by defining in/out local distribution and the theoretical analysis. We suggest that FL can be interpreted as balancing between attaining new knowledge from local distribution while preserving old knowledge for the global distribution.
>
> Despite its simplicity, the proposed FedNTD is an effective algorithm to balance the trade-off between those two components, as discarding true-class logits conceptually decouples those two components. Thereby, FedNTD controls their trade-off by a single hyperparameter $\beta$, which we use 1.0 in our paper.
>
> We claim that our novelty lies not just in the proposed method but in interpreting the forgetting view in FL and following theoretical analysis to support its behavior and analyze the effect of the proposed method.

---

> ### Comment · Reviewer_sAmP · 2022-08-08
> **Thank the authors for their detailed response**
>
> Thank the authors for their detailed response. After reading the author feedback, Weakness 2 and Weakness 4 have been addressed. However, I still have the following concerns/suggestions:
>
> 1. The current theoretical analysis cannot clearly explain the extent to which FedNTD can alleviate forgetting. Can you provide a brief overview of the analysis in this paper?
> 2. I ran the source code, but I did not find that not-true distillation has observable advantages in my experimental setting: Take five domains of DomainNet (http://ai.bu.edu/DomainNet/) dataset as five clients, in each round, the number of local epoch is 1, the sample ratio is 1, each domain has 345 classes. In the first five global rounds, KD and NTD have completely consistent performance. Does it indicate that NTD has limited effects when there are a large number of categories?
> 3. The results of the baseline methods are questionable. 1) With 100 clients and sample ratio 0.1, FedAvg should have at least 80% Acc on MNIST dataset. 2) FedProx is very sensitive to hyperparameters. By adjusting $\mu$, it should have much higher performance than FedAvg, especially in the case of non-IID.

---

> > ### Author Response · Authors · 2022-08-09
> > **Q3) The results of the baseline methods**
> >
> > **1) Performance of FedAvg on MNIST**
> >
> > In our experiments, Cutout[1] augmentation is used for the MNIST (as specified in our supplementary material **Appendix C.2** Datasets). It makes the FL problem more challenging without affecting the heterogeneity level as its randomness is identical across the clients. We believe that this is why the reported FedAvg is against the reviewer’s intuition. Here we provide the accuracy of the MNIST dataset *with (w/)* and *without (w/o)* Cutout by varying heterogeneity levels.
> >
> > | **(w/o Cutout)** | **LDA (a=0.05)** | **LDA (a=0.1)** | **LDA (a=0.5)** | **Sharding (s=2)** | **Sharding (s=5)** | **Sharding (s=10)** |  **iid** |
> > |:----------:|:----------------:|:---------------:|:---------------:|:------------------:|:------------------:|:--------------------:|:--------:|
> > | **FedAvg** |       84.1       |       96.0      |       97.9      |        95.8        |        97.5        |         **98.1**         |   **98.4**   |
> > | **FedNTD** |     **90.2**     |     **96.2**    |     **98.0**    |      **96.7**      |      **97.6**      |       **98.1**       | **98.4** |
> >
> > **Table1: MNIST test accuracy (w/o Cutout)**
> >
> >
> >
> > | **(w/ Cutout)** | **LDA (a=0.05)** | **LDA (a=0.1)** | **LDA (a=0.5)** | **Sharding (s=2)** | **Sharding (s=5)** | **Sharding (s=10)** |  **iid** |
> > |:----------:|:----------------:|:---------------:|:---------------:|:------------------:|:------------------:|:--------------------:|:--------:|
> > | **FedAvg** |       39.9       |       79.7      |       87.7      |        78.6        |        85.1        |         88.3         |   **89.5**   |
> > | **FedNTD** |     **57.8**     |     **81.3**    |     **87.9**    |      **84.4**      |      **86.6**      |       **89.1**       | **89.5** |
> >
> > **Table2: MNIST test accuracy (w/ Cutout)**
> >
> > &nbsp;
> >
> > --------------
> >
> > &nbsp;
> >
> > **2) Performance of FedProx on Non-IID Setups**
> >
> > As the reviewer mentioned, FedProx[2] is very sensitive to its hyperparameter of the proximal term. For FedProx, we chose the hyperparameter that performs **best** from the candidate set {0.1, 0.5, 1.0}. The hyperparameters for the other algorithms are provided in our supplementary material **Appendix C.4**: Algorithm Implementation Details. Although FedProx is a prevalent algorithm to mitigate heterogeneity in FL, its performance gain over FedAvg is often negligible [3,4,5,6,7] or even degrades sometimes [3, 6, 7].
> >
> > According to the literature we surveyed, the gain from the FedProx tends to be dominant when the the amount of local data samples is severely different across clients, and a small fraction of clients participates in each round. We could observe a similar tendency in our work (**Table 1** in **Section 4.2** and **Appendix H**. Additional Experiments in our supplementary material).
> >
> > In our experiments, we validate the proposed algorithm on both distribution skewness and quantity skewness by using two different data partitioning strategies: (i) *Sharding Strategy (distribution skewness)* and (ii) *LDA Strategy: (distribution & quantity skewness)*.
> >
> >
> > [1]. [DeVries, Terrance, and Graham W. Taylor. "Improved regularization of convolutional neural networks with cutout." arXiv preprint arXiv:1708.04552 (2017).](https://arxiv.org/abs/1708.04552)
> >
> > [2]. [Li, Tian, et al. "Federated optimization in heterogeneous networks." Proceedings of Machine Learning and Systems 2 (2020): 429-450.](https://proceedings.mlsys.org/paper/2020/hash/38af86134b65d0f10fe33d30dd76442e-Abstract.html)
> >
> > [3]. [Li, Qinbin, et al. "Federated learning on non-iid data silos: An experimental study." 2022 IEEE 38th International Conference on Data Engineering (ICDE). IEEE, 2022.](https://ieeexplore.ieee.org/abstract/document/9835537)
> >
> > [4]. [Mendieta, Matias, et al. "Local Learning Matters: Rethinking Data Heterogeneity in Federated Learning." Proceedings of the IEEE/CVF Conference on Computer Vision and Pattern Recognition. 2022.](https://openaccess.thecvf.com/content/CVPR2022/html/Mendieta_Local_Learning_Matters_Rethinking_Data_Heterogeneity_in_Federated_Learning_CVPR_2022_paper.html)
> >
> > [5]. [Li, Qinbin, Bingsheng He, and Dawn Song. "Model-contrastive federated learning." Proceedings of the IEEE/CVF Conference on Computer Vision and Pattern Recognition. 2021.](https://openaccess.thecvf.com/content/CVPR2021/html/Li_Model-Contrastive_Federated_Learning_CVPR_2021_paper.html)
> >
> > [6]. [Collins, Liam, et al. "Exploiting shared representations for personalized federated learning." International Conference on Machine Learning. PMLR, 2021.](https://scholar.google.com/scholar?hl=en&as_sdt=0%2C5&q=Exploiting+Shared+Representations+for+Personalized+Federated+Learning&btnG=)
> >
> > [7]. [Li, Xin-Chun, et al. "Federated Learning with Position-Aware Neurons." Proceedings of the IEEE/CVF Conference on Computer Vision and Pattern Recognition. 2022.](https://openaccess.thecvf.com/content/CVPR2022/html/Li_Federated_Learning_With_Position-Aware_Neurons_CVPR_2022_paper.html)

---

> > ### Author Response · Authors · 2022-08-09
> > **Q2) Gain of FedNTD in DomainNet experiment**
> >
> > Most of all, we sincerely appreciate your effort to validate our work in the various setups. In our work, we assume that the class distribution of local datasets differs across clients in heterogeneous FL scenarios and define forgetting as the local model’s accuracy drop in the classes that the global model predicts well. To compensate for it, we define out-local distribution $\tilde{P}(D)$ in our **Definition 1** and propose NTD to preserve global knowledge on it.
> >
> > In the suggested DomainNet experiment, the source of heterogeneity is not in class distribution (*label skewness*) but in the class-conditional distribution (*feature skewness*), in which even the features from the same class differ across clients. In this case, the class-wise forgetting is insignificant, and the defined out-local distribution does not sufficiently reflect the skewness of local distribution from the global distribution, limiting the gain of NTD over KD.
> >
> > In addition, although we fixed the hyperparameter for FedNTD in our work, a smaller temperature $\tau$ is recommended when the dataset has many classes. As NTD exploits the not-true class logits only, the difference between logit scales is small; thereby, the probability should be more *sharpened*.
> >
> > We would like to mention that there is another line of work that investigates the heterogeneity induced by explicitly different source domains across clients [1- 7], which are primarily considered in the Personalized FL context [3,4,5] or with Medical Imaging [6,7]. To the best of our knowledge, the Domain Generalization in a generic FL setting is still an open question.
> >
> > [1]. [Chen, Hong-You, and Wei-Lun Chao. "On bridging generic and personalized federated learning for image classification." International Conference on Learning Representations. 2021.](https://openreview.net/forum?id=I1hQbx10Kxn)
> >
> > [2]. [Luo, Zhengquan, et al. "Disentangled Federated Learning for Tackling Attributes Skew via Invariant Aggregation and Diversity Transferring." arXiv preprint arXiv:2206.06818 (2022).](https://arxiv.org/abs/2206.06818)
> >
> > [3]. [Li, Xiaoxiao, et al. "Fedbn: Federated learning on non-iid features via local batch normalization." arXiv preprint arXiv:2102.07623 (2021).](https://arxiv.org/abs/2102.07623)
> >
> > [4]. [Sun, Benyuan, et al. "Partialfed: Cross-domain personalized federated learning via partial initialization." Advances in Neural Information Processing Systems 34 (2021): 23309-23320.](https://proceedings.neurips.cc/paper/2021/hash/c429429bf1f2af051f2021dc92a8ebea-Abstract.html)
> >
> > [5]. [Huang, Wenke, Mang Ye, and Bo Du. "Learn From Others and Be Yourself in Heterogeneous Federated Learning." Proceedings of the IEEE/CVF Conference on Computer Vision and Pattern Recognition. 2022.](https://openaccess.thecvf.com/content/CVPR2022/html/Huang_Learn_From_Others_and_Be_Yourself_in_Heterogeneous_Federated_Learning_CVPR_2022_paper.html)
> >
> > [6]. [Yao, Chun-Han, et al. "Federated multi-target domain adaptation." Proceedings of the IEEE/CVF Winter Conference on Applications of Computer Vision. 2022.](https://openaccess.thecvf.com/content/WACV2022/html/Yao_Federated_Multi-Target_Domain_Adaptation_WACV_2022_paper.html)
> >
> > [7]. [FedDG: Federated Domain Generalization on Medical Image Segmentation via Episodic Learning in Continuous Frequency Space](https://openaccess.thecvf.com/content/CVPR2021/html/Liu_FedDG_Federated_Domain_Generalization_on_Medical_Image_Segmentation_via_Episodic_CVPR_2021_paper.html)
> >
> > [8]. [Jiang, Meirui, Zirui Wang, and Qi Dou. "Harmofl: Harmonizing local and global drifts in federated learning on heterogeneous medical images." Proceedings of the AAAI Conference on Artificial Intelligence. Vol. 36. No. 1. 2022.](https://ojs.aaai.org/index.php/AAAI/article/view/19993)

---

> > ### Author Response · Authors · 2022-08-09
> > **Q1) Overview of the theoretical analysis**
> >
> > Our theoretical analysis aims to provide the importance of knowledge preservation on out-local distribution and how NTD achieves it.
> >
> > >In **Proposition 1**, we suggest that increasing the regularization coefficient on out-local distribution quadratically reduces the local gradient diversity in the FL system, guiding them towards the global gradient. We also note that if we use the uniform weighting instead of out-local distribution, we get a strictly bigger dissimilarity since the variance will not be reduced (**Equation 24** in **Appendix Q**. Proof of Proposition 1 in our supplementary material)
> >
> > >In **Proposition 2**, we suggest that NTD is the regularization method on the out-local distribution by showing that it can be expressed as the weighted loss sum on the out-local distribution.
> >
> > >In **Proposition 3**, we suggest that the global model, which predicts each class evenly well, reduces the expected learning distance, which we defined as the distance between global model parameters and learned local model parameters. This explains how the knowledge preservation by NTD propagates its benefits across communication rounds.
> >
> > The theoretical analysis of the *extent* of forgetting is very hard since we need to consider such as the following in advance:
> >
> > - The extent of increase of loss on out-local distribution when fitting on the in-local distribution
> > - The interaction between the KL-Div loss using global prediction and the CE Loss using the ground-truth label
> > - The noisy prediction of global model in the intermediate communication rounds
> >
> > However, analysis under such strict assumptions may not provide useful insights which practically hold.
> >
> > In practice, a possible extension of our work would be adaptively changing the regularization coefficient across datasets, clients, or learning phases in local training/communication rounds, as the extent of forgetting can be empirically measured by using loss or accuracy.

---

### Official Review · Reviewer_eF2S · 2022-07-11

**Rating:** 7
**Confidence:** 4
**Soundness:** 4 excellent
**Presentation:** 3 good
**Contribution:** 4 excellent

**Summary:**

This paper addresses the data heterogeneity problem in federated learning, by drawing an analogy with the forgetting problem in continual learning. The authors find that the global model forgets the knowledge from previous rounds, when aggregating clients' locally trained models that optimize toward the local data distribution. They propose a novel and simple algorithm to mitigate the forgetting of knowledge. Specifically, the authors add to the training loss a term that penalizes the discrepancy of the predicted class distribution between the local model and the global model, where the discrepancy considers all but the ground-truth class. The authors perform experiments to thoroughly compare with state-of-the-art federated learning methods and demonstrate the preservation of global knowledge in their method.

**Questions:**

The authors may respond to the comments raised in the "Strengths And Weaknesses" section.


**Limitations:**

The authors are aware that privacy is an important subject in federated learning and state that their approach does not compromise data privacy. The authors also alert that bias in the global model, if any, may cause a similar tendency in the local models.


**Strengths And Weaknesses:**


The proposed method is simple and elegant. It appears to work quite well. The authors study the heterogeneity problem in federated learning by drawing inspiration from continual learning and knowledge distillation. They successfully develop a technique that can cope with data heterogeneity. The idea is intuitive and it could be easily used by practitioners.

The authors spend a significant amount of effort to study the problem of forgetting. They define a forgetting measure, illustrate the forgetting behavior, justify suppressing gradient diversity as a way to mitigate forgetting, propose a novel loss term for training, and demonstrate its effectiveness in preserving the global knowledge. All in all, the work is solid and well done.

The paper is generally well written, but it needs thorough proofreading. Many obvious grammatical errors are spotted all over the paper.

A few comments for improvement:

- Line 123. It is a bit unclear what "... (the) accuracy varies on $p(D)$ and $\tilde{p}(D)$" means. The authors may want to elaborate.

- Line 146. It is unclear, in the first pass of reading, why the local gradient $\nabla f^k$ is computed in the said way. This may have something to do with the loss function (10) that is proposed only later. The authors may want to give some motivation before Proposition 1.

---

> ### Author Response · Authors · 2022-08-02
> **Improved clarity with proofreading**
>
> We revised the paper's grammatical errors aided by professional service, which will be further thoroughly revised on the camera-ready version. For the comments for improvement, we rewrite and add the expressions as follows:
>
> - **Line 123. "... (the) accuracy varies on p(D) and p~(D)"**
>
> For clarity, we rewrite the expression as follows:
> > *“... the change of global and local models’ accuracy on p(D) and p~(D) during each communication round …”*.
>
>
> - **Line 146. “... the local gradient ∇fk = (pk + βp~k) · g…”**
>
> As pointed out, we intend to deliver the idea of how the information on the out-local distribution modifies local gradients and aligns them with motivation to develop a loss function that adequately guides local gradient toward the global data distribution.
>
> We express the local gradient as the linear combination of two gradient components weighted on (i) in-local distribution $p_k$ and (ii) out-local distribution $\tilde{p}_k$. In general, β equals 0 as the local gradient is obtained only from the locally available data distribution. To clarify it, we add some motivation before **Proposition 1** as follows:
>
>
> >*“...To understand the effect of preserving knowledge on the out-local distribution $\tilde{p}(D)$, we analyze how the local gradients and their diversity varies by adding gradient signal on $\tilde{p}(D)$ with factor β and obtain the following proposition. …”*

---

### Author Response · Authors · 2022-08-08
**The end of Reviewer-Author Discussions phase is approaching**

Dear Reviewers,

Could you please go over our responses and the revision since we can have interactions with you only by this Tuesday (9th)? We have responded to your comments and faithfully reflected them in the revision, and provided additional experimental results and discussions that address the suggested concerns.

We sincerely thank you for your time and efforts in reviewing our paper and for your insightful and constructive comments.

Thanks, Authors

---

### Meta-Review · Area_Chair_nh9p · 2022-08-26

**Recommendation:** Accept
**Confidence:** Less certain

**Metareview:**

This paper studies the data heterogeneity problem in federated learning. It is well-written, with some novel ideas and good efforts on analyzing the problem of forgetting and experimental studies. We hope the authors can revise the paper carefully per the reviewers' suggestion and add some new experimental results during rebuttal into the final version.

**Award:**

No

---

### Decision · Program_Chairs · 2022-09-14

Accept